# Metabolic modelling as a powerful tool to identify critical components of *Pneumocystis* growth medium

Olga A. Nev[1,2]*, Elena Zamaraeva[3], Romain De Oliveira[4], Ilia Ryzhkov[5], Lucian Duvenage[6,7], Wassim Abou-Jaoudé[4], Djomangan Adama Ouattara[4], Jennifer Claire Hoving[6,7], Ivana Gudelj[1], Alistair J. P. Brown[1,2]

**1** Department of Biosciences, Faculty of Health and Life Sciences, University of Exeter, Exeter, United Kingdom, **2** Medical Research Council Centre for Medical Mycology at the University of Exeter, Exeter, United Kingdom, **3** Leverhulme Research Centre for Functional Materials Design, Materials Innovation Factory, University of Liverpool, Liverpool, United Kingdom, **4** Gencovery, Lyon, France, **5** DBAX, Exeter, United Kingdom, **6** CMM AFRICA Medical Mycology Research Unit, Institute of Infectious Diseases and Molecular Medicine (IDM), **7** Department of Pathology, Faculty of Health Sciences, University of Cape Town, Cape Town, South Africa

* o.nev@exeter.ac.uk

**Data Availability Statement:** KEGG metabolic maps can be found at: https://figshare.com/s/3a260486d5d8a6067b16 All the necessary files

## Abstract

Establishing suitable *in vitro* culture conditions for microorganisms is crucial for dissecting their biology and empowering potential applications. However, a significant number of bacterial and fungal species, including *Pneumocystis jirovecii*, remain unculturable, hampering research efforts. *P. jirovecii* is a deadly pathogen of humans that causes life-threatening pneumonia in immunocompromised individuals and transplant patients. Despite the major impact of *Pneumocystis* on human health, limited progress has been made in dissecting the pathobiology of this fungus. This is largely due to the fact that its experimental dissection has been constrained by the inability to culture the organism *in vitro*. We present a comprehensive *in silico* genome-scale metabolic model of *Pneumocystis* growth and metabolism, to identify metabolic requirements and imbalances that hinder growth *in vitro*. We utilise recently published genome data and available information in the literature as well as bioinformatics and software tools to develop and validate the model. In addition, we employ relaxed Flux Balance Analysis and Reinforcement Learning approaches to make predictions regarding metabolic fluxes and to identify critical components of the *Pneumocystis* growth medium. Our findings offer insights into the biology of *Pneumocystis* and provide a novel strategy to overcome the longstanding challenge of culturing this pathogen *in vitro*.

## Author summary

*Pneumocystis jirovecii* is a human pathogen that causes life-threatening pneumonia in hundreds of thousands of immunocompromised individuals each year. Neither this fungus nor its close relative, the mouse pathogen *Pneumocystis murina*, can be cultured *in vitro*, and this is significantly hindering scientific progress. Therefore, we developed a

and instructions to reproduce the results from the paper can be found in the following GitHub repository: https://github.com/nevolga/optmed.

**Funding:** O.A.N. was the recipient of a Skills Development Fellowship from the MRC (MR/V006169/1) https://www.ukri.org/councils/mrc/ and an NIHR BRC Exeter Translational Fellowship (NIHR203320) https://www.exeterbrc.nihr.ac.uk/. A.J.P.B. was supported by grant from Wellcome (224323/Z/21/Z) https://wellcome.org/ and the MRC Programme Grant (MR/M026663/2) https://www.ukri.org/councils/mrc/. E.Z. acknowledges funding from the Leverhulme Trust via the Leverhulme Research Centre for Functional Materials Design (RC-2015-036) https://www.leverhulme.ac.uk/. L.D. acknowledges funding from the Carnegie Corporation of New York (CCNY) https://www.carnegie.org/. J.C.H. was supported by grant from Wellcome (209293) https://wellcome.org/. O.A.N., A.J.P.B., L.D., and J.C.H. acknowledge funding from the MRC Centre for Medical Mycology at the University of Exeter (MR/N006364/2 and MR/V033417/1) https://www.ukri.org/councils/mrc/ and the NIHR Exeter Biomedical Research Centre https://www.exeterbrc.nihr.ac.uk/. The views expressed are those of the author(s) and not necessarily those of the NIHR or the Department of Health and Social Care. The funders had no role in study design, data collection and analysis, decision to publish, or preparation of the manuscript.

**Competing interests:** The authors have declared that no competing interests exist.

comprehensive genome-scale metabolic model for *P. murina* using bioinformatics, software tools, and recently published genome data, and we used this metabolic model to predict critical components required for growth of the fungus. Our findings suggest that a subset of amino acids and specific lipids are essential for *Pneumocystis* survival. Additionally, we employed non-classical Flux Balance Analysis and Reinforcement Learning approaches to optimise ingredients for a *Pneumocystis* growth medium. This novel methodology has provided new insights into *Pneumocystis* metabolism and offers a potential approach to overcoming the challenge of culturing this pathogen *in vitro*, which would accelerate progress towards the development of improved diagnostics and therapies.

## Introduction

The experimental dissection of microorganisms, such as bacteria and fungi, is greatly facilitated through access to suitable *in vitro* culture conditions that enable their independent growth within a laboratory setting. This dissection enhances our understanding of these microbes, their biology, and the potential applications of these organisms. By cultivating bacteria and fungi, we are able to investigate mechanisms that underlie the infectious diseases they elicit, refine diagnostic assays for these diseases, and develop novel vaccines and therapeutic interventions.

A significant number of bacterial and fungal species remain unculturable, thereby restricting research into these organisms. Examples of unculturable bacteria include the segmented filamentous bacteria, which are frequently found in the gut of mammals and are thought to modulate the immune system and promote gut-associated lymphoid tissue development [1]. Additionally, unculturable *Oscillospira* species in the human gut are thought to be important for fibre degradation, immune modulation, and metabolic health [2,3]. There is even a list of the "most wanted" taxa from the human microbiome, which includes numerous bacterial species that are considered highly significant for human health but remain unculturable [4]. Viable but non-culturable (VBNC) cells represent another group of bacteria that is challenging to culture. These cells are metabolically active but cannot form colonies on standard culture media [5]. Over 80 bacterial species are currently known to belong to this group including members of the Proteobacteria and, in particular, the human pathogen *Vibrio parahaemolyticus* [6]. In the fungal kingdom, the cultivation of obligate intracellular Microsporidial pathogens is not fully standardised in most research laboratories, which hampers the improvement of diagnostic techniques as well as investigations of their pathogenesis [7]. Furthermore, the metabolism and biology of fungi in the *Malassezia* genus are poorly understood because of difficulties in culturing them, despite their importance for human health. These fungi have been associated with skin disorders that include dandruff and seborrheic dermatitis [8].

Another striking example of a non-culturable fungus is the human pathogen *Pneumocystis jirovecii*. This fungus colonises the lungs of infants [9,10] and causes life-threatening pneumonia in immunocompromised individuals and transplant patients [11,12]. *Pneumocystis* remains one of the most common and serious infections in HIV/AIDS patients [13] and is a particularly serious problem for developing countries, such as those in Sub-Saharan Africa. The fungus is also an emerging issue in non-HIV patients in developed countries, such as UK [14] and USA [15]. Although *Pneumocystis* has a major impact on human health, substantial gaps persist in our understanding of the biology and epidemiology of this opportunistic pathogen. The inability to culture *Pneumocystis in vitro*, despite three decades of research, makes this pathogen uniquely difficult to study. There are reports describing conditions that support

the axenic growth of *Pneumocystis* [16,17], but others have found these difficult to replicate [18,19]. The lack of well-established *in vitro* culture methods has been recognised as *the* major obstacle in *Pneumocystis* research. So far, attempts to overcome bioinformatically predicted auxotrophic requirements by simple supplementation of growth media has not been sufficient to solve the problem and achieve long-term growth *in vitro*, though several groups have reported limited short-term growth [19]. Therefore, we have employed metabolic modelling as an alternative approach to address this major bottleneck in *Pneumocystis* research.

To identify metabolic requirements and imbalances that prevent the growth of this major fungal pathogen *in vitro*, we have developed, for the first time, a comprehensive *in silico* genome-scale metabolic model (GEM) of *Pneumocystis* growth and metabolism. Our study focuses on the mouse-specific fungal pathogen, *Pneumocystis murina* with a view to empowering, in the future, experimental comparisons with growth in murine models of infection. Genomic comparisons suggest only minor metabolic differences between *P. murina* and the human and rat pathogens, *P. jirovecii* and *P. carinii* [20]. GEMs are mathematical representations of an organism's metabolic reactions based on genome annotation data and experimentally obtained information [21,22]. GEMs are often used in combination with Flux Balance Analysis (FBA) [23], which is a computational method to predict metabolic fluxes across the network to achieve a certain optimisation objective. Potential objectives include maximising growth or reducing resource consumption. This powerful approach has been successfully applied to metabolic engineering, strain development, drug discovery, enzyme functional predictions, phenotypic characterisations, interspecies interactions modelling and the understanding of human diseases [24–26]. Furthermore, metabolic modelling has been used to design and optimise the composition of culture media. For instance, GEMs have been successfully applied to formulate *in vitro* growth media for bacteria such as *Lactobacillus plantarum* [27] and *Bacteroides caccae* [28]. In addition, GEMs have been used to design minimal growth media for the lactic acid bacterium *Lactococcus lactis* [29], a potentially beneficial gut microbe *Faecalibacterium prausnitzii* [30], and for the human pathogens *Staphylococcus aureus* [31], *Neisseria meningitidis* [32], and *Helicobacter pylori* [33].

In recent years, the development of GEMs for fungi has significantly advanced our understanding of fungal metabolism and its application in various biotechnological and medical fields. Key examples include GEMs for filamentous fungi [34], industrially important fungi [35], anaerobic gut fungi [36], human pathogens [37,38] and yeasts [39–41].

In this paper we describe the development of a GEM reconstruction for *Pneumocystis* metabolism. We review and refine the current *Pneumocystis* genome annotation using available information in the literature together with well-established bioinformatics tools, and then organise these data into a well-defined GEM network. We then use literature-based assumptions and available software tools to describe a process of biomass production by defining a biomass equation. To validate the model, we utilise *in vitro* growth culture medium conditions designed by Merali *et al* [16], which have been reported to support *Pneumocystis* growth. By extending this with relaxed Flux Balance Analysis (FBA), we show how the model can be used to make important predictions regarding metabolic flux to identify critical components of the *Pneumocystis* growth medium. Finally, we propose how to optimise Merali's culture medium composition by using a Reinforcement Learning approach.

## Results

### Metabolic model development

To develop a metabolic model for *Pneumocystis*, we followed a well-established protocol for generating a high-quality genome-scale metabolic reconstruction proposed by Thiele and

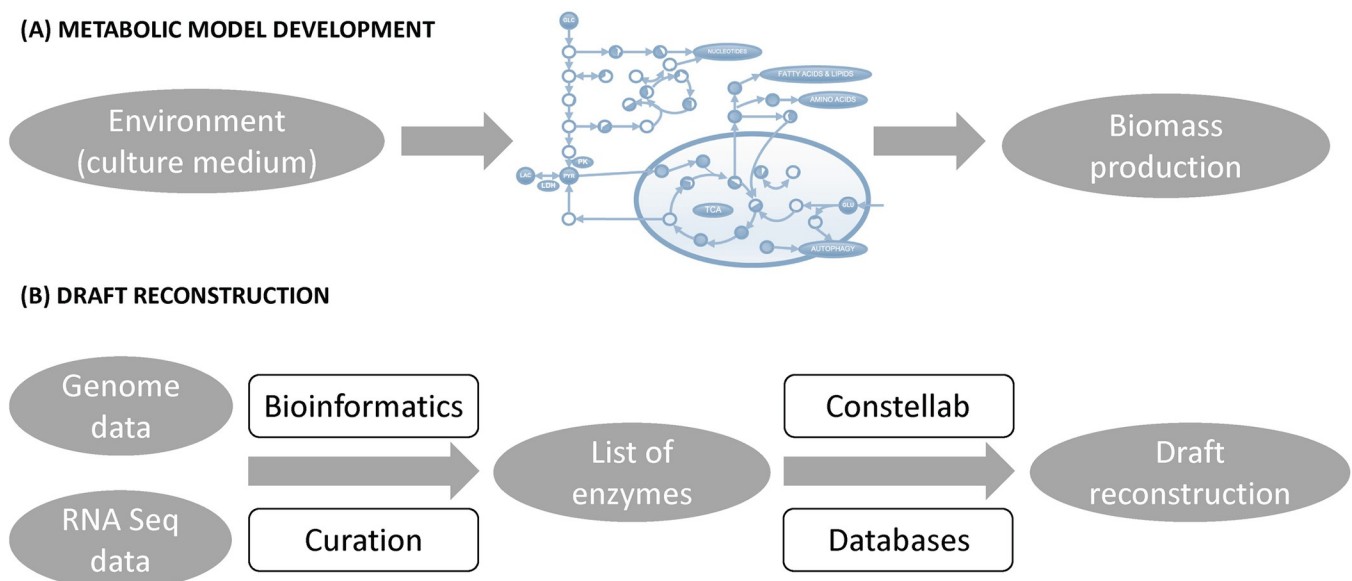

**Fig 1. Schematic representation of the metabolic model development and draft reconstruction processes. (A)** The metabolic model development process includes the reconstruction of a draft network, formulation of a biomass equation, and validation of the resulting model using experimental data on the organism's phenotypes under specified environmental conditions. **(B)** To create a high-quality draft reconstruction for *Pneumocystis*, the latest genome annotation was utilised, reviewed, and refined using bioinformatics tools and the literature. This information was further combined with RNA-Seq data and manually curated to create the list of functional enzymes present in *Pneumocystis*. From this list, the draft reconstruction was generated using the *Gencovery* software Constellab and the CHEBI, BRENDA, and RHEA databases.

Palsson [22]. The procedure generally consists of the following key steps. Firstly, a draft reconstruction is created. Next, a biomass equation is defined. Finally, the reconstruction is translated into a mathematical format, and the resulting model is verified, evaluated and validated using published information about the organism's phenotypes under various environmental conditions. A schematic representation of this process is shown in Fig 1A.

## Draft reconstruction

To generate a high-quality draft reconstruction for *Pneumocystis*, we used the most recent version of the genomic sequence [20]. In order to gain a more accurate picture of the metabolites and nutrient transport functions in *Pneumocystis*, we reviewed these data and, where necessary, refined them using available information in the literature including recently published RNA-Seq data [42] (see *Materials and Methods*). Finally, to achieve the most accurate results possible, we performed re-evaluation and manual curation of the entire draft reconstruction and the associated network content. We then used the brick GENA (Genome-Based Network Analysis) of Constellab platform (see *Materials and Methods*) to organise these data into a well-defined genome-scale metabolic network for *Pneumocystis* using CHEBI [43], BRENDA [44], and RHEA [45] databases to define gene-protein-reaction associations. The schematic representation of the draft reconstruction process is shown in Fig 1B.

As a result, we obtained a network consisting of 487 chemical reactions corresponding to 495 enzymes thought to operate in a *Pneumocystis* cell. We compared *Pneumocystis* metabolism with that of the model yeasts *S. pombe* and *S. cerevisiae* using the KEGG Mapping Tool [46] (Fig 2A). This metabolic map clearly shows that *P. murina* has a reduced genome that lacks metabolic pathways essential in other yeasts [20]. In particular, *Pneumocystis* has partially lost the gluconeogenesis pathway and glyoxylate cycle and is incapable of synthesising all 20

**(A) GLOBAL METABOLIC PATHWAYS**                          **(B) ARGININE BIOSYNTHESIS**

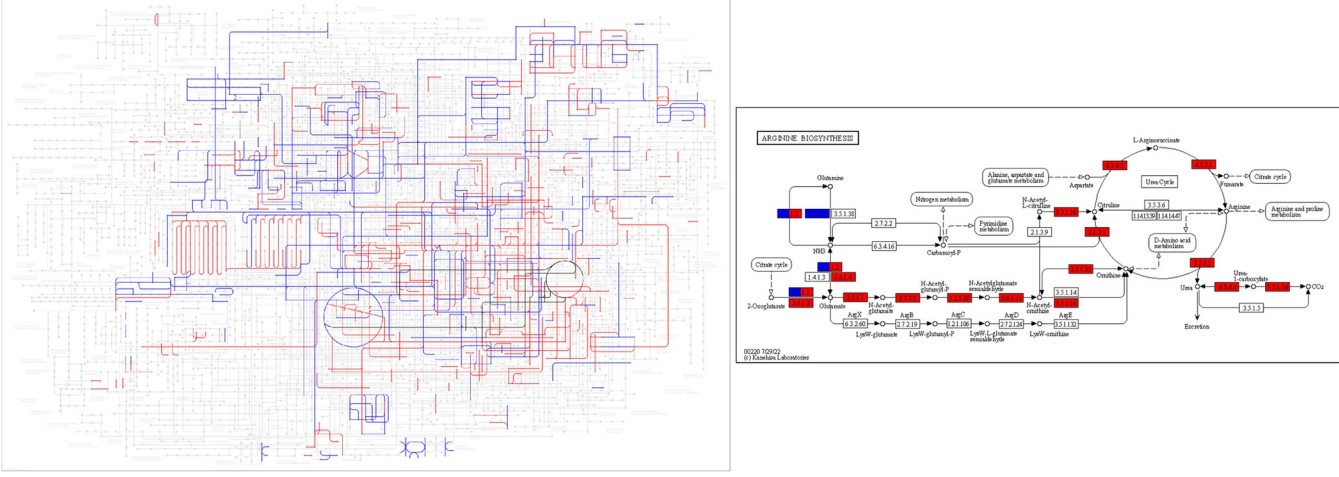

present in *Pneumocystis*

present in model yeasts (*S. pombe* & *S. cerevisiae*), but not in *Pneumocystis*

**Fig 2. *Pneumocystis* metabolism in comparison with the model yeasts. (A)** Comparison of the *Pneumocystis* metabolic network with reactions present in the model yeasts *S. pombe* and/or *S. cerevisiae* using the KEGG Mapping Tool. Metabolic pathways present in *Pneumocystis* are highlighted in blue, whereas those pathways present in model yeasts, but not in *Pneumocystis*, are featured in red. The arginine biosynthetic pathway, highlighted in (B), is shown in black. **(B)** Comparison of arginine metabolism in *Pneumocystis* and the model yeasts *S. pombe* and *S. cerevisiae* using the KEGG Mapping Tool. Enzymes present in *Pneumocystis* are highlighted in blue, and those present in model yeasts are in red.

essential amino acids from scratch, as well as some polyamines, lipids, vitamins, and cofactors [20,47]. For example, as shown in Fig 2B, the fungus lacks the enzymes required to synthesise a crucial arginine precursor, ornithine, from glutamate and to convert it to citrulline. Furthermore, the production of arginine from aspartate is also blocked due to the lack of the requisite enzymes. Other yeasts possess these enzymes and, therefore, are capable of arginine biosynthesis. Interestingly, according to this metabolic map, *Pneumocystis* can convert glutamine into glutamate and vice versa. Hence, auxotrophy for certain specific amino acids may be circumvented via supplementation by others. The metabolic pathways that have been lost or retained in *Pneumocystis* are summarised in Table 1.

## Biomass equation

A biomass equation accounts for all known biomass constituents and their fractional contributions to the overall biomass of a cell [48]. The biomass reaction is of utmost significance for a model's analysis as it provides an indication of whether growth is increasing or decreasing in response to specific environmental conditions being studied.

To begin with, the main course-grained components of the biomass are considered and then broken down into smaller elements. Typically, the main biomass precursors include proteins, carbohydrates, lipids, RNA, and DNA as well as small molecules, such as cofactors. These macro components can be further broken down into their micro constituents. For example, proteins are essentially a sum of amino acids, whereas DNA and RNA consist of nucleotides, and lipids can be present as a pool of fatty acids and/or sterols.

Ideally, a detailed biomass composition is determined experimentally for the organism of interest. However, due to the inability to grow *Pneumocystis in vitro*, it was not feasible to

**Table 1. Summary of metabolic pathways lost or retained in *Pneumocystis*.**

| PATHWAY | LOST | RETAINED | PARTLY LOST |
|---|---|---|---|
| **Nucleotide metabolism** | Nucleotide salvage pathway | Nucleotide *de novo* synthesis | |
| **Carbohydrate metabolism** | Nucleotide sugar biosynthesis from galactose<br>Pyruvate fermentation<br>Chitin and mannan biosynthesis<br>Beta-glucan biosynthesis and degradation (troph form only)<br>Glucuronate pathway | Glycolysis<br>TCA cycle<br>Pentose phosphate pathway (oxidative part)<br>PRPP biosynthesis<br>Glycogen degradation<br>Nucleotide sugar biosynthesis from glucose<br>UDP-N-acetyl-D-glucosamine biosynthesis<br>Conversion of fructose and mannose to glucose<br>Trehalose utilisation<br>Beta-glucan biosynthesis and degradation (cyst form only) | Pentose phosphate pathway (non-oxidative part)<br>Gluconeogenesis<br>Inositol phosphate metabolism<br>Glycogen biosynthesis<br>Glyoxylation |
| **Lipid metabolism** | Fatty acids degradation<br>Conversion of fecosterol and episterol to ergosterol<br>Myo-inositol, choline, complex sphingolipids, ether lipids, phosphatidylinositol, phosphatidylcholine *de novo* synthesis<br>Cytosolic pathway for fatty acids<br>Synthesis of glycerol from glycerone-phosphate or monoacylglycerol | Fatty acid biosynthesis and elongation in mitochondria and ER<br>Fecosterol and episterol synthesis<br>Alternative mechanisms to supply cells with phosphatidylinositol, phosphatidylcholine, inositol, choline | Fatty acid biosynthesis, initiation and elongation<br>Fatty acids beta-oxidation<br>Synthesis of ergosterol<br>Synthesis of cholesterol |
| **Cofactor metabolism** | Pantothenate *de novo* synthesis<br>Thiamine, biotin, ubiquinone, siderophores *de novo* synthesis<br>Reductive iron assimilation<br>Synthesis of CoA | Conversion of pantothenate to CoA<br>NAD *de novo* synthesis and its salvage<br>Heme biosynthesis<br>Riboflavin biosynthesis | |
| **Amino acid metabolism** | 20 amino acids *de novo synthesis*<br>Polyamine *de novo* synthesis | Cysteine biosynthesis from homocysteine + serine | Shikimate pathway<br>Glutathione biosynthesis from glutamate |
| **Energy metabolism** | Sulfur metabolism<br>V-type ATPase | Cytochrome bc1 complex<br>Succinate dehydrogenase<br>Carbon fixation | Formaldehyde assimilation<br>NADH dehydrogenase<br>Cytochrome c oxidase<br>F-type ATPase |
| **Glycan metabolism** | | N-glycan precursor biosynthesis<br>O-glycan biosynthesis | N-glycan precursor trimming<br>N-glycosylation by oligosaccharyltransferase<br>N-glycan biosynthesis<br>GPI-anchor biosynthesis |
| **Endocytosis** | | Clathrin-dependent endocytosis | |
| **CO2 hydration** | whole pathway | | |
| **Stress response pathway** | pH sensing<br>Osmotic stress response<br>Oxidative stress response<br>Cell wall stress response | | |

obtain a detailed biomass composition for this fungus. Therefore, we estimated the relative fractions of each biomass precursor using genome data, software tools, pre-existing metabolic models of species closely related to *Pneumocystis*, and the limited information about the pathogen's composition available in the literature.

In particular, we estimated *Pneumocystis* protein, DNA, and RNA content from genomic data [20] with the aid of a biomass tool developed by Santos & Rocha [49]. This tool calculates the microbial biomass composition in amino acids and nucleotides based on genomic and transcriptomic data, accepting input files in FASTA format and transcriptomic data in CSV format. It is reported to serve as a viable alternative in the absence of experimental data [49–51]. To account for the lipid content, we relied on available experimental data reported on the

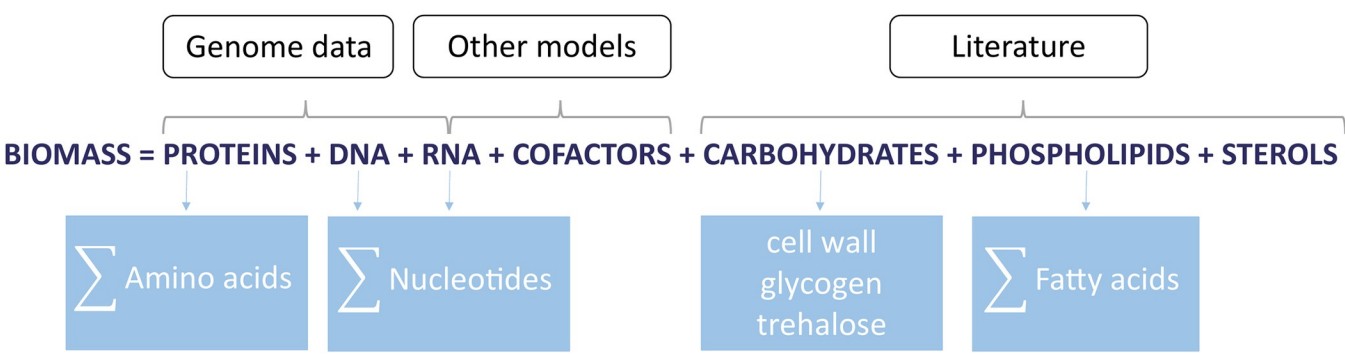

**Fig 3. Schematic representation of the biomass equation.** It is assumed that the biomass of *Pneumocystis* consists of the following macro-nutrients: proteins, DNA, RNA, carbohydrates, phospholipids, sterols, and cofactors. Some of these are made up of smaller micro-nutrients that include amino acids, nucleotides, and fatty acids. The contribution of each biomass precursor in *Pneumocystis* is estimated by using genome data, software tools, metabolic models of species closely related to *Pneumocystis*, and available experimental data on *Pneumocystis* composition in the literature.

average composition of phospholipids and sterols in *Pneumocystis* [52–54]. However, due to the unavailability of experimental data, the majority of the content for the soluble pool of small metabolites was derived from a comprehensive general list as suggested by Thiele and Palsson [22] (S1 File for more details). Other assumptions were drawn from the biomass content of *S. pombe* and its energy requirements for polymerisation (ATP) [41] combined with available experimental data on *Pneumocystis* cell wall composition [55,56] as well as the presence of polyamines and carbohydrates in the cell [47,57].

The schematic representation of the biomass equation is shown in Fig 3. Detailed information on the process of the biomass reaction reconstruction can be found in S1 File.

## Flux balance analysis

Flux Balance Analysis (FBA) [23] is typically used to calculate and predict metabolic fluxes in a constructed genome-scale metabolic network. This powerful approach has been extensively applied in various physiological and biological studies [24]. For example, FBA has been successfully used to identify and resolve "knowledge gaps" in metabolic networks [58], to study the impact of drugs in pathogens and to identify potential drug targets [59,60], to model host-pathogen interactions [61], to simulate multiple gene knockouts [62], to predict the yield of important cofactors [63], and to support metabolic engineering [64–66].

To apply FBA and calculate metabolic fluxes within the network, a metabolic network reconstruction must first be developed. This reconstruction includes a list of stoichiometrically balanced biochemical reactions which need to be converted into a mathematical model. To this end, a numerical matrix $S$ of a size ($m{\times}n$) is constructed, where $m$ rows represent metabolites $X_i$ (with $i{\in}\{1...m\}$) and $n$ columns represent reactions $j$ (with $j{\in}\{1...n\}$), and metabolic fluxes $v_j$ (with $j{\in}\{1...n\}$) are assigned to the corresponding reactions (Fig 4). The matrix $S$ consists of stoichiometric coefficients from the biochemical reactions, with positive numbers representing metabolites produced in the reactions, negative numbers representing metabolites consumed in the reactions, and zero representing metabolites that do not participate in the reactions. With a help of matrix $S$ the system of biochemical reactions can then be rewritten as a system of differential equations describing how metabolite concentrations $X$ change over time in the network: $\frac{dX}{dt} = S \times v$.

Classical FBA assumes that the metabolic system is in a steady state, where metabolite production and consumption are balanced. In the current work, this assumption was adapted to a quasi-steady state with relaxed boundary conditions (referred to as "relaxed Flux Balance

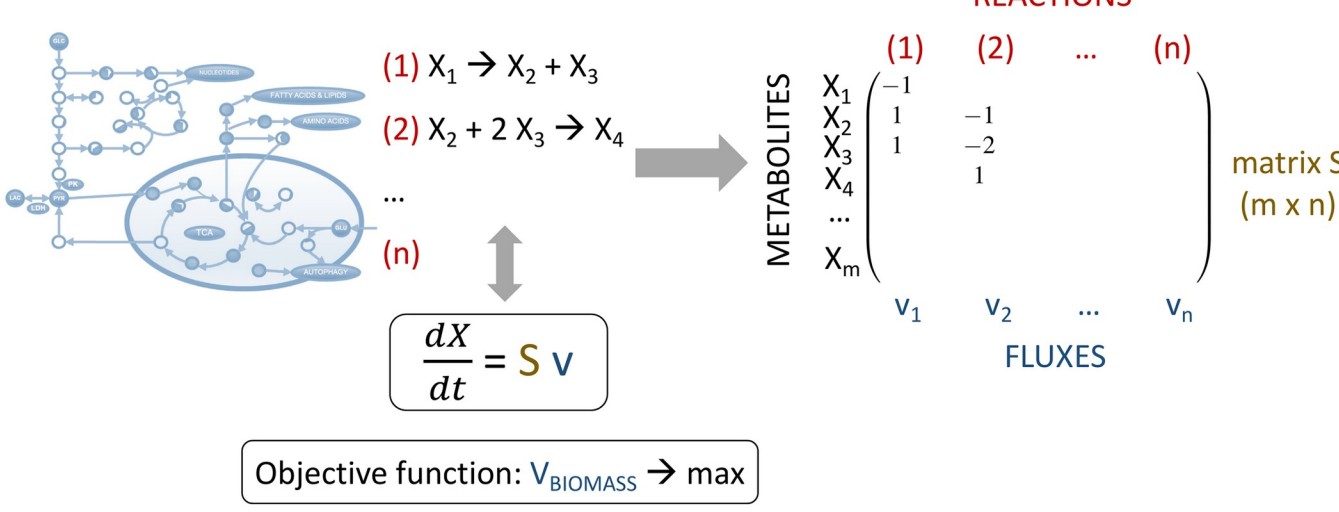

**Fig 4. Schematic representation of the FBA approach.** To implement the FBA approach, biochemical reactions are converted into a mathematical model (see *Biomass equation* for details). Classical FBA relies on the assumption that the metabolic system is in a steady state, but in this study this assumption was relaxed by assuming that certain intermediate metabolites may not reach steady state (see text). Mathematically, biomass production as a biological objective is formulated as a function that maximises the biomass flux, which is equivalent to the exponential growth rate of the organism.

Analysis" by Fleming *et al* [67]) to enable feasible model simulation and subsequent analyses (see *Materials and Methods*).

Typically, constraints are used to reduce the FBA solution space. These constraints rely on biological knowledge or experiment data on the organism. For instance, experimental data can be used to derive intake/excretion fluxes and thus constraints on extracellular fluxes. However, the inability to cultivate *Pneumocystis in vitro* has led to a deficiency of essential empirical information. Thus, we are mostly dependent on the reconstruction, open data collection, and curation processes used to contextualise the model before optimisation.

Since we are interested in predicting growth in *Pneumocystis*, our biological objective is biomass production. We mathematically define this objective as the objective function of the biomass flux $v_{BIOMASS}$ and assume that this flux is equivalent to the organism's exponential growth rate. We employed a quadratic programming method with a relaxed quasi-steady-state assumption (QSSA) to identify a particular flux distribution in the network that maximises the biomass objective function (see *Materials and Methods*). Fig 4 provides a schematic representation of the FBA approach we applied.

## Model validation

A genome-scale metabolic model can be verified, validated, and evaluated using available experimental data [22]. To ensure that the model is free of technical errors, the model is usually tested against known constraints. Under normal circumstances, when growth data are available for the target organism, to test whether the model accurately represents the metabolism of the organism, the predictions of the model are compared with experimental data obtained under various environmental conditions. Finally, to check whether the model is reliable and applicable for the intended purposes, one can perform simulations using the model and compare the results with the actual biological outcomes. Then the model can be refined and iterated to improve its accuracy and usefulness. These steps help to ensure that the genome-scale metabolic model is a reliable tool for predicting and analysing the metabolism of the organism under various conditions.

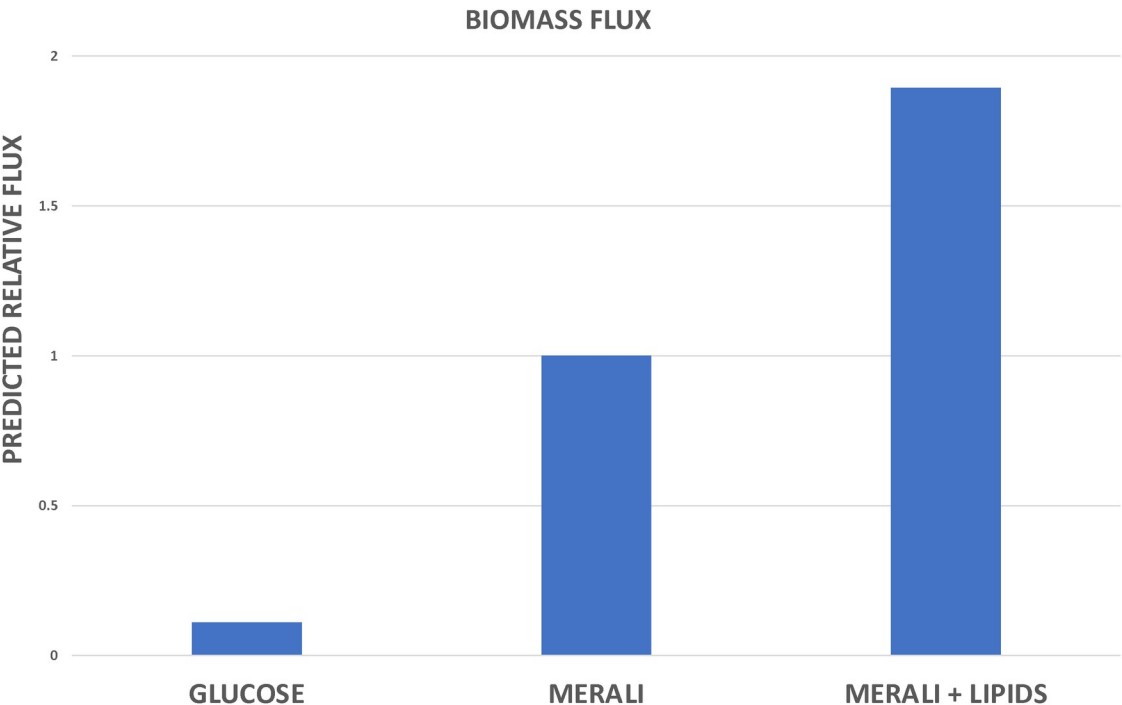

**Fig 5. Metabolic model validation and predictions.** The biomass flux predicted by the model under the following conditions: only glucose is present; growth medium designed by Merali *et al* [16]; and Merali's medium supplemented with the additional lipids present in the *Pneumocystis* biomass equation. Biomass flux is expressed relative to that observed for Merali's medium.

Due to the lack of a well-established *in vitro* culture medium for *Pneumocystis*, experimental data are limited and often obtained under *ex vivo* conditions (i.e. growth with cultured mammalian cells) which are not suitable for the model's validation. To validate our metabolic model, we employed *in vitro* growth culture medium conditions designed by Merali *et al* [16], as described in *Materials and Methods*. We compared the biomass flux values obtained by the model's simulations under Merali's conditions with those calculated for the culture medium where only glucose is present. Genetic data suggest that *Pneumocystis* relies largely on glucose utilisation via oxidative pathways for energy production [47], but cannot grow on this nutrient alone. Fig 5 shows that the model predicts a higher biomass flux under Merali's conditions, as expected, as compared to glucose alone.

## Model predictions

The model of *Pneumocystis* growth and metabolism can be exploited to analyse the composition of Merali's medium and to identify its most significant components for the growth of *Pneumocystis*.

### *In silico* drop off and gene knockout experiments

To identify the most important amino acids for *Pneumocystis* growth present in Merali's medium, we assessed their contributions to biomass flux by sequentially removing each amino acid individually from the growth medium and calculating the corresponding biomass flux. The results suggest that not all amino acids contribute equally to *Pneumocystis* growth. In particular, the model predicts that a deficiency of isoleucine or histidine in the medium results in a significant reduction in biomass flux, whereas arginine or tyrosine deficiencies do not (Fig

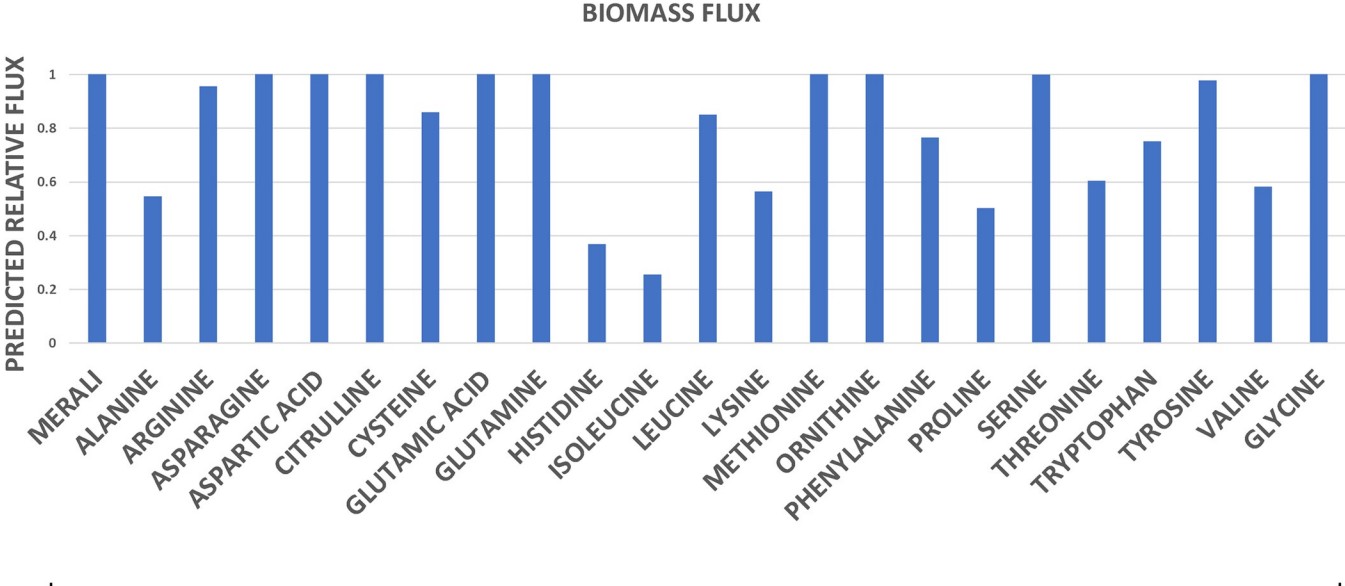

**Fig 6. *In silico* drop off experiment: amino acids.** The biomass flux predicted by the model in the medium designed by Merali *et al* [16]; and in the same medium but where each amino acid is individually omitted. Biomass flux is expressed relative to that observed for Merali's medium.

6). One possible explanation may be the differential contributions of particular amino acids to the overall biomass (see *Biomass Equation* for more details). Indeed, according to estimates of protein content from genomic data using a java application [49], both isoleucine and histidine contribute a relatively high weight per gram of biomass, whereas arginine and tyrosine contribute the least to biomass (S1 File).

This reasoning was based on the respective weights of amino acids in the biomass equation and may not be applicable to certain amino acids, such as serine and glycine, for example. These amino acids contribute approximately as much to the biomass equation as isoleucine and histidine, and yet their absence from the growth medium is not predicted to decrease the biomass flux (Fig 6). This observation is consistent with the idea that *Pneumocystis* is capable of synthesising certain amino acids from others. In particular, *Pneumocystis* can interconvert serine and glycine, indicating that the elimination of glycine from the growth medium may not dramatically affect the biomass flux, since the pathogen may still be able to synthesise glycine from serine or *vice versa*. To investigate this hypothesis, we employed our model to simulate an *in silico* gene knockout experiment. The enzyme serine hydroxymethyltransferase (SHMT, EC 2.1.2.1) catalyses the interconversion of serine and glycine. Therefore, the gene encoding SHMT was conceptually "switched off", to simulate a typical gene knockout experiment in the laboratory. The model predicts that the biomass flux remains relatively unchanged for wild type cells under the medium conditions designed by Merali and co-workers, even when serine, glycine, or both are omitted from the medium (Fig 7A). However, when the SHMT enzyme is "switched off" via a conceptual gene knockout, and neither glycine nor serine are available, the biomass flux drops due to the inability of *Pneumocystis* to interconvert serine into glycine. Essentially, this *in silico* experiment corresponds to the inhibition of serine-glycine interconversion as well as the exclusion of both amino acids from the medium (Fig 7B). Since serine and glycine contribute significantly to the amino acid content of *Pneumocystis* biomass, their elimination is predicted to lead to a decrease in the biomass production.

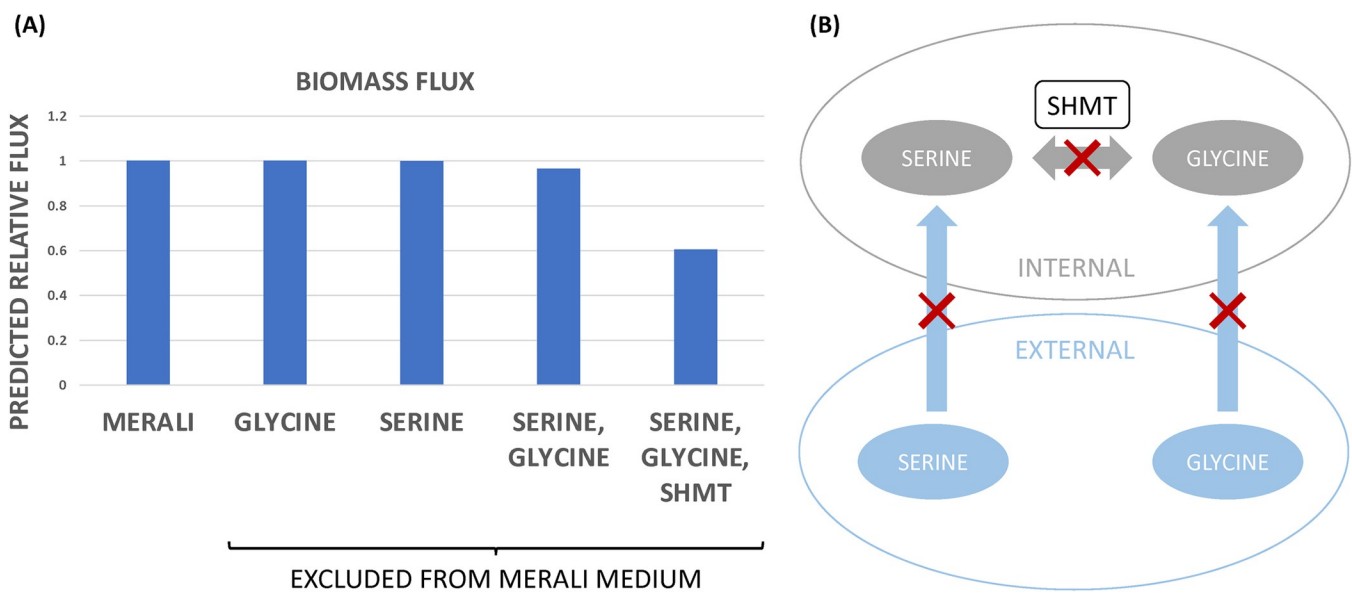

**Fig 7. *In silico* gene knockout experiment: serine and glycine interconversion. (A)** The biomass flux predicted by the model in the medium designed by Merali *et al* [16] and in equivalent medium lacking serine, glycine, or both, or where the SHMT enzyme is "switched off". Biomass flux is expressed relative to that observed for Merali's medium. **(B)** Schematic representation of the above *in silico* experiment.

## Critical role of lipids in the *Pneumocystis* growth

To identify the key lipids in Merali's medium that are crucial for *Pneumocystis* growth, we evaluated their impact on biomass flux using an analogous approach. Not all lipids appear to contribute equally to *Pneumocystis* growth. Fig 8 shows that cholesterol and palmitic, stearic,

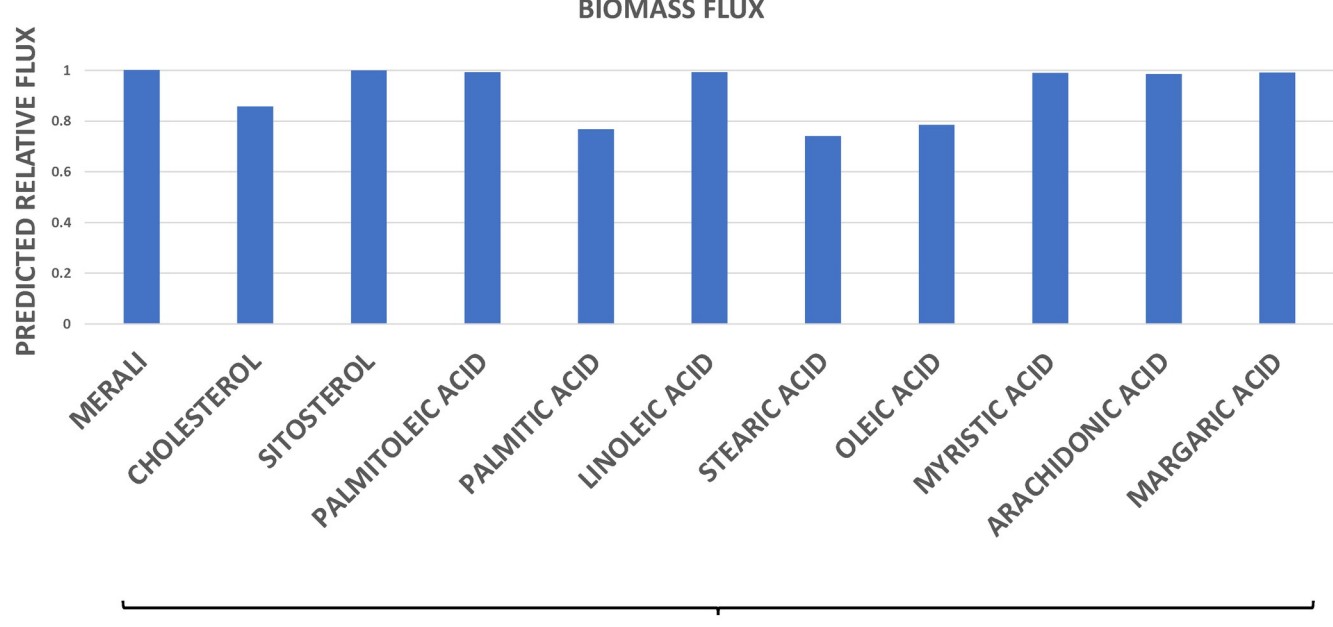

**Fig 8. *In silico* drop off experiment: lipids.** The biomass flux predicted by the model in the medium designed by Merali *et al* [16] and in Merali's medium lacking each individual lipid. Biomass flux is expressed relative to that observed for Merali's medium.

and oleic acids are the most important lipids for *Pneumocystis* growth present in Merali's medium. Indeed, experimental studies showed that cholesterol was the most abundant sterol in the *Pneumocystis* cell membrane [52,65], whereas palmitic, stearic, and oleic fatty acids were the dominant fatty acids [52]. Moreover, it has been demonstrated experimentally that the pathogen is able to assimilate cholesterol and these fatty acids from the medium to generate phospholipids and form new membranes [65,66].

However, certain sterols and fatty acids, which have been experimentally identified in *Pneumocystis* [52–54], and therefore included in the biomass reaction, were not included by Merali and co-workers in their growth medium [16]. This discovery led us to formulate the hypothesis that supplementing the medium with extra lipids might enhance *Pneumocystis* growth. We employed our metabolic model to test this hypothesis by simulating the effects of supplementing Merali's medium with additional lipids upon the resultant biomass flux. The model predicted that supplementation with additional lipids would significantly increase the biomass flux compared to previously observed values, suggesting that these components may enhance *Pneumocystis* growth (Fig 5).

## Growth medium optimisation

Our model represents a valuable tool to make predictions regarding growth medium composition and can be used to design new culture media or to refine existing growth conditions. Here, we demonstrate how the model can facilitate the optimisation of Merali's medium. In particular, we illustrate how the number of components in Merali's medium might be reduced while maintaining equivalent biomass flux values, without compromising growth performance.

To predict the minimal set of nutrients that can produce the maximal possible biomass flux value, the irreducible subset(s) of the initial nutrient set must be examined. This is the subset that contains those nutrients that cannot be removed without reducing the biomass flux value. Due to the large number of nutrients in Merali's medium (155) [16], it is impractical to explore every possible nutrient combination. Hence, alternative heuristic approaches are needed. Furthermore, specific nutrients may exhibit synergistic effects, implying that the optimal minimal subset size is not only determined by the initial nutrient set size, but also by the order in which individual nutrients are removed. As a result, a simple exclusion of nutrients from the initial list would not yield the optimal solution, necessitating a more sophisticated approach.

Reinforcement Learning (RL) has demonstrated its effectiveness in solving optimisation problems where brute force methods are impractical or computationally infeasible [67]. For instance, this robust technique has been applied to various chemical problems [68] including crystal structure prediction [69], used successfully in drug discovery [70], personalised treatment development and clinical decision making [71,72], and applied to social and behavioural neurosciences [73].

By exploiting RL to optimise the composition of Merali's growth medium (see *Materials and Methods*), we were able to decrease the initial set of 155 nutrients by about 76% to only 37 nutrients while maintaining the biomass flux value. After 300,000 iterations of training the RL algorithm we applied successfully identified and removed non-essential nutrients (S2 Fig). The algorithm predicted 815 minimal sets of nutrients of size 37, where 58 of these sets were unique. Interestingly, 29 core nutrients were the same in all minimal sets: the other 8 nutrients came from a list of 19 non-core nutrients that varied from set to set (Table B in S4 File). Table 2 shows the list of 37 nutrients that appeared most frequently, with a total of 649 occurrences.

We also verified the flux values computed by the metabolic model under Merali's medium conditions and discovered 91 zero fluxes associated with extracellular transport reactions

**Table 2. Most frequently occurring optimal minimal 37-nutrient set predicted by the Reinforcement Learning algorithm.**

| Nutrient | Core/Non-core nutrient |
|---|---|
| Alanine | core |
| Arginine | core |
| Cysteine | core |
| Histidine | core |
| Isoleucine | core |
| Leucine | core |
| Lysine | core |
| Phenylalanine | core |
| Proline | core |
| Threonine | core |
| Tryptophan | core |
| Tyrosine | core |
| Valine | core |
| Arachidonic acid | core |
| Linoleic acid | core |
| Margaric acid | core |
| Myristic acid | core |
| Oleic acid | core |
| Palmitic acid | core |
| Palmitoleic acid | core |
| Stearic acid | core |
| Cholesterol | core |
| Sitosterol | core |
| Folate | core |
| Pantothenate | core |
| Protoheme | core |
| Putrescine | core |
| Riboflavin | core |
| Thiamine | core |
| Asparagine | non-core |
| Glycine | non-core |
| Glutamate | non-core |
| Water | non-core |
| Hydrogenphosphate | non-core |
| Pyridoxal | non-core |
| S-adenosyl-L-methionine | non-core |
| Methionine | non-core |

(Table A in S4 File). These fluxes do not appear to contribute to biomass production and, therefore, the corresponding ambient nutrients from the medium can be excluded. We compared the nutrients identified as non-essential for growth by the RL approach with nutrients whose corresponding flux values were zero. The RL algorithm identified a significantly greater number of non-essential nutrients, considerably simplifying the original 155-component list of Merali's medium composition to only 37 nutrients.

By using this task as an example, we showed that our novel RL approach can be successfully applied to make further predictions regarding growth medium composition as well as to design new culture media or to refine existing growth conditions.

## Model predictions and empirical data

Significantly, the results obtained by applying RL reinforce our previous findings (see *Model predictions*, above) by showing that crucial amino acids and lipids from Merali's growth medium are present in the final optimal minimal nutrient set predicted by the RL algorithm (Table 2). Significantly, the model's predictions are consistent with the empirical data available in the literature.

## Experimental observations

The RL approach identified cholesterol, as well as palmitic, stearic, and oleic fatty acids, as essential core components of Merali's medium required for *Pneumocystis* growth. Indeed, it was shown experimentally that these are dominant lipids in the composition of the pathogen [52,65] and that *Pneumocystis* cells are able to assimilate cholesterol and these fatty acids from the environment [65,66].

Moreover, Merali and co-workers reported that *Pneumocystis* growth was observed when p-aminobenzoic acid, putrescine, and protoheme were omitted individually, but that their addition enhanced growth [16]. Our RL approach identified these components of Merali's medium as essential, classifying putrescine and protoheme as core nutrients and p-aminobenzoic acid as non-core, which is consistent with Merali's observations.

## The SAM dilemma

S-Adenosyl-L-methionine (SAM) is a co-substrate that is involved in various metabolic processes, playing crucial roles in methylation reactions and polyamine biosynthesis. SAM is synthesised from L-methionine and ATP via the enzyme SAM synthase.

Merali *et al* [16] did not experimentally detect any activity of the SAM synthase and thus suggested that *P. carinii* is a SAM auxotroph [74]. They also reported that the inclusion of SAM in the growth medium enhances *Pneumocystis* growth and that it is absolutely required for continuous growth of the fungus [16]. Furthermore, a *Pneumocystis* gene with homology to mitochondrial SAM transporters was discovered, which supported the hypothesis that the fungus scavenges this critical nutrient from its host [75]. However, others have suggested that a gene encoding SAM synthase is present in *Pneumocystis* [20]. Furthermore, this gene was cloned, and its enzymatic activity was confirmed via expression in *Escherichia coli* [76]. These results suggest that *Pneumocystis* may not depend on its host for the supply of this important cofactor.

We used the model to explore this interesting conundrum relating to SAM biosynthesis and metabolism in *Pneumocystis* [77]. Interestingly, the model predicted that both SAM and L-methionine are essential, but non-core components of Merali's medium. The optimal nutrient sets identified by the RL contained either SAM or L-methionine or both. This supports the view that, based on their metabolic interactions in *Pneumocystis* (Fig 9), the fungus requires either methionine or SAM for growth *in vitro*.

## Discussion

We have developed, for the first time, a genome-scale metabolic model for *Pneumocystis*. In order to do so, we reviewed and improved the current *Pneumocystis* genome annotation by incorporating relevant information from the literature and utilising established bioinformatics tools and then organised this information into a well-defined GEM network. We used literature assumptions and available software tools to generate a biomass equation and validated our metabolic model by utilising *in vitro* growth culture medium conditions proposed by

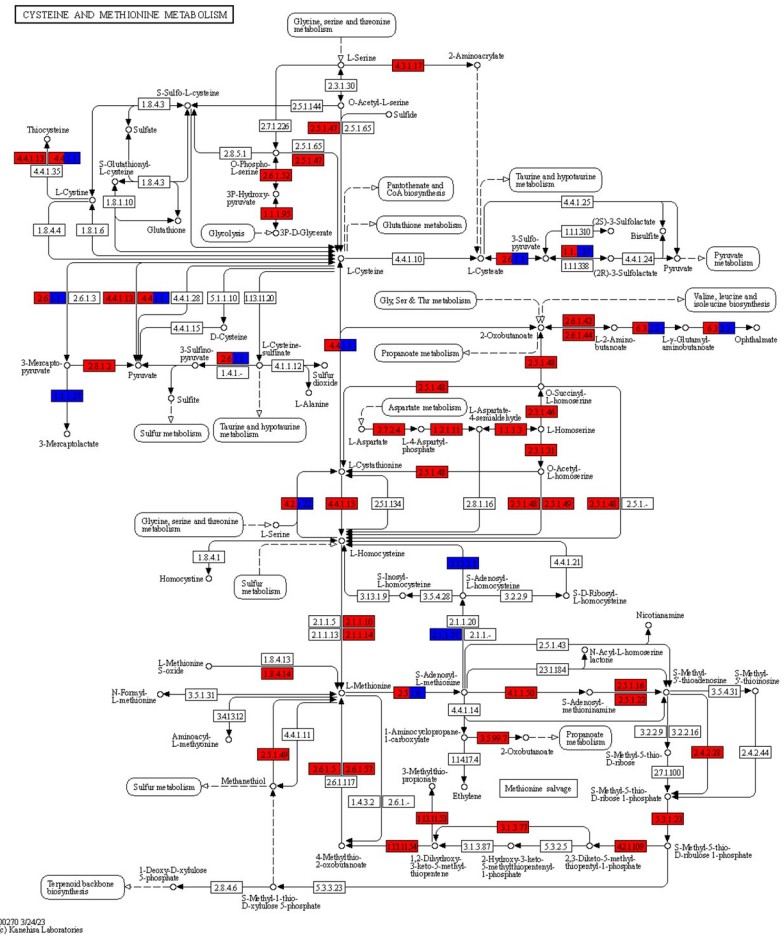

**Fig 9. The SAM dilemma.** Comparison of cysteine and methionine metabolism in *Pneumocystis* and in *S. pombe* and *S. cerevisiae* using the KEGG Mapping Tool. Enzymes present in *Pneumocystis* are highlighted in blue, while those present in model yeasts are in red.

Merali *et al* [16]. By employing a relaxed Flux Balance Analysis (FBA) approach, we calculated biomass flux values to confirm that our metabolic model is a powerful tool to investigate biological questions that cannot be fully explored through conventional bioinformatics and experimental approaches. Moreover, using the model, we made novel predictions regarding *Pneumocystis* growth and metabolism. In particular, we analysed Merali's growth medium and identified its critical components as well as suggested how to supplement it to enhance *Pneumocystis* growth. We have also demonstrated that our model can replicate drop-off and gene knockout experiments, highlighting its potential as a valuable tool in the development of new antifungal drug targets. Specifically, our model can be used to predict the likely impact of inhibiting specific metabolic targets upon fungal growth. Finally, by employing a Reinforcement Learning approach, we propose how to simplify Merali's medium to achieve a minimal growth culture medium.

Reconstructing metabolic networks can be a challenging process for organisms with incomplete genome annotations, low sequence homology to other organisms and/or unique pathways [78]. As a result, incomplete reconstructions may occur and the quality of the constraints used to build the model may be compromised, which limits the efficacy and precision of the

FBA approach [79,80]. The remarkable nature of *Pneumocystis*, with its extremely reduced genome and dependence on the host for both nutrients and a stable environment [47], requires special attention in the model development process. Since the pathogen possesses certain nutrients transporters and retains an endocytosis pathway, it has the potential to scavenge nutrients from the host [20]. This implies that the metabolic network of *Pneumocystis* may exhibit genuine gaps that reflect the pathogen's reliance on its host. These gaps should not be filled automatically during the reconstruction process, as this could lead to inaccurate predictions. The FBA approach, however, can be used to predict how these "actual" metabolic gaps may be complemented to promote the growth of the fungus *in vitro*. In addition to the supplementation of nutritional requirements that have been predicted bioinformatically, some of which have been tested already [19], FBA is capable of predicting how nutrient supplementation might be applied to address metabolic imbalances created, for example by loss of energetic or redox homeostasis. This will help to identify the growth phenotypes of *Pneumocystis* under specified media conditions and suggest ways to supplement the nutrient requirements of the pathogen.

The FBA approach, although well-established, has limitations [81]. For example, it cannot predict nutrient concentrations, since it does not incorporate kinetic parameters. Additionally, in its classical form, FBA is only applicable to determine fluxes at a steady state. Moreover, FBA does not take account of regulatory effects such as enzyme activation by protein kinases or gene regulation, nor does it consider stress response or signalling pathways, and so its predictions may not always be accurate. To improve the accuracy of FBA predictions, the integration of regulatory and signalling network information into the metabolic model is likely to be useful. This approach represents a valuable extension to the classical FBA approach and has been previously employed to obtain a more detailed and dynamic picture of biological networks [82,83].

In this study, we used a relaxed QSSA FBA approach to analyse the contribution of certain metabolites to biomass production in *Pneumocystis*. While this approach provides valuable insights into those nutrients likely to yield optimal metabolic fluxes, it does not directly predict auxotrophies. Therefore, this methodology has limitations in identifying specific auxotrophies. Further experimental work will be required to identify *Pneumocystis*' precise nutrient dependencies.

Experimental data play a crucial role at each stage of developing and refining genome-scale metabolic models [81]. To construct and refine metabolic networks, experimental data on the presence or absence of enzymes, metabolites, and reactions can be integrated with transcriptomic and proteomic data. The construction of the biomass equation, as well as setting constraints for metabolic fluxes, also require experimental data. Furthermore, to validate the model's predictions, a comparison with experimental data is necessary. This creates a feedback loop between the model and the experiments, allowing for the identification of errors and gaps in the model, and informing and refining the metabolic network. However, in the case of *Pneumocystis*, the inability to grow this pathogen *in vitro* has resulted in a lack of critical empirical data on its composition, physiological and biological characteristics, kinetics of nutrient uptake, and enzymatic activities. The absence of experimental data has hindered the development and refinement of a comprehensive metabolic model for *Pneumocystis*, highlighting the importance of experimental data in the process.

For example, data on enzyme specificity regarding substrate and cofactor usage is exclusively available in organism-specific biochemical literature or databases. Lack of such information for *Pneumocystis* compelled us to rely on databases that are not organism-specific, such as RHEA [45] and BRENDA [44]. Additionally, acquiring data on *Pneumocystis*-specific gene and reaction localisation proved to be challenging. Hence, we assumed that the proteins reside

in the cytoplasm. Moreover, due to the deficiency of critical experimental data, we considered reactions in our initial draft reconstruction reversible, despite the fact that the directionality of reactions may have significant impact on the model's performance. It is worth noting that the reconstruction proposed in this study is subject to revision and refinement in the subsequent versions of the *Pneumocystis* metabolic model, as more biochemical data become available.

The hypotheses generated here by the model need to be tested experimentally. Moreover, the growth of *Pneumocystis in vitro* should be compared experimentally against its growth *in vivo* in the host (as an important control). Clearly, it is not possible to test this in humans. Therefore, for practical and ethical reasons, we developed a genome-scale metabolic model for the mouse-specific fungal pathogen, *Pneumocystis murina*. However, this platform allows us to generate analogous models for other host-specific pathogens, which will provide the basis for the future development of *in vitro* culture conditions for each of these species. Recent genome sequencing has confirmed that the *Pneumocystis* genus includes several species, each of which is highly specific to a particular mammalian host [84]: human-specific *Pneumocystis jirovecii*, mouse-specific *Pneumocystis murina*, rat-specific *Pneumocystis carinii* and *Pneumocystis wakefieldiae*, rabbit-specific *Pneumocystis oryctolagi*, macaque-specific *Pneumocystis macacae*, and dog-specific *Pneumocystis canis*. Their genome sequences have been recently reported [20,85–87]. Metabolic models developed for these *Pneumocystis* species can also be linked to the corresponding metabolic models for their specific mammalian hosts. This will enhance the understanding of the metabolic relationships between these pathogens and their host, permit detailed comparison of the metabolic gaps for each of these pathogens, and potentially provide invaluable clues about the potential basis for their species-specificity.

The lack of *in vitro* culture methods has been recognised as the major obstacle in *Pneumocystis* research. For example, the diagnosis of *Pneumocystis* infections relies primarily on microscopic detection in respiratory specimens [47]. Using a powerful combination of *in silico* metabolic modelling and experimental phenotyping opens new opportunities to release this major experimental bottleneck. This is extremely important because a reliable *in vitro* culture method will empower the dissection of *Pneumocystis* immunology and pathobiology [88,89], leading, in the longer-term, to the development of new antifungal therapies that circumvent the emerging resistance of *Pneumocystis* to azole drugs [20,90], and to dramatic advances in *Pneumocystis* biology and therapy.

## Materials and methods

### Bioinformatics tools

In this study, for practical and ethical reasons, we developed a genome-scale metabolic model for the mouse-specific fungal pathogen, *Pneumocystis murina*. To review and refine the genome annotation for *P. murina*, we used well-established bioinformatics tools to compare this pathogen with other species that are relatively closely related to it in phylogenetic terms [91]. In particular, we selected *Schizosaccharomyces pombe* and *Saccharomyces cerevisiae* as comparison organisms due to the availability of functional annotations for these species.

To identify homologous genes between these organisms and *P. murina*, we used BLAST [92] in combination with the UniProtKB sequence database [93] (BLASTp with default parameters, filtering alignment at coverage of 80%, identity/similarity of 80%, and p-value $> = 0.05$) and keeping only fungal identified proteins to avoid potential false positive results. Protein sequences for *P. murina* (PRJNA70803) were downloaded from NCBI as deposited by Ma *et al* [20]. The latest version of protein sequences for *S. cerevisiae* S288c (GCA_0001460452) and *S. pombe* (GCA_000002945) were downloaded from the *Saccharomyces* Genome Database [94] and *PomBase* Genome Database [95], respectively.

We then used PFAM [96] (with default parameters, filtering results for E-value $> = 0.05$) to further annotate functional domains in previously selected genes to better link them to gene ontology (GO) terms and enzyme commission (EC) numbers [97,98].

Finally we checked the obtained list of enzymes against the biochemical reaction database BRENDA [44] and RNA Seq data [42], and also used available, albeit limited, experimental data on *Pneumocystis* to curate enzymes manually.

This enabled the generation of a comprehensive list of 495 EC numbers corresponding to the enzymes present in *P. murina*, which provided a starting point to build a metabolic reconstruction network.

## Constellab software tool

To build a metabolic reconstruction network based on the 495 EC numbers that correspond to the enzymes found in *P. murina*, we used a Constellab platform (configuration gws_core: 0.7.2, gws_gena: 0.6.3, gws_biota: 0.6.1, exe_gena: 0.2.0, gws_omix: 0.6.1), which was developed by *Gencovery*. This company specialises in metabolic modelling and offers state-of-the-art expertise in this field [99]. Constellab is a cloud-based open bioinformatic platform that enables the efficient analysis of complex biological datasets, such as large-scale omics data. In the context of this study, Constellab allowed us to create digital twins of cellular metabolism [100], which are digital representations of the metabolic processes that can be used to simulate, analyse, and optimise their performance and efficiency under various situations. In this way, the Constellab platform allowed us to create a metabolic model for *P. murina*, which served as a digital twin of its metabolism and growth.

To achieve this goal, we established gene-protein-reaction associations using the *Gencovery*'s in-house BIOTA open database, which is a comprehensive and structured repository of 'omics data sourced from the open European EMBL-EBI database [101] and the NCBI taxonomy database [102]. The BIOTA database also contains manually curated ontology data as well as molecular data on taxonomy, enzymes, metabolites, and metabolic pathways from comprehensive databases, such as CHEBI [43], BRENDA [44], and RHEA [45]. By mapping our data to the BIOTA, we were able to build an initial version of the *P. murina* metabolic network, which consisted of 528 chemical reactions that corresponded to the 495 identified enzymes (see *Bioinformatics tools*, above). The mapping was performed using algorithms available in the open brick GENA.

## Manual curation

The initial automated draft reconstruction, comprising 528 reactions, was subjected to additional manual curation (S2 File). As a result, five additional reactions were manually incorporated by drawing information from the KEGG [46] and RHEA [45] databases, as well as from the *S. cerevisiae* model [40] (Table A in S2 File).

We then assessed whether the network was balanced in terms of mass and charge and identified 52 reactions displaying imbalances in mass and/or charge. Most of these imbalanced reactions contained chemical compounds in a generic form ($(1{\rightarrow}4)$-$\alpha$-D-glucan being represented as $(1,4$-$\alpha$-D-glucosyl$)_n$, for example), making it impossible to calculate their mass or charge correctly. Such generic reactions are considered non-specific due to insufficient knowledge and biochemical evidence, and hence were typically excluded from the network [22]. As a result, 44 imbalanced reactions were deleted. The other 8 reactions were kept due to their retention in the *S. cerevisiae* [40] and *S. pombe* models [41] as well as their importance according to the literature and likely contribution to the biomass production. The stoichiometry of all reactions was sourced from the RHEA database [45] (Table B in S2 File).

Next, we eliminated the "isolates" from the reconstruction, i.e. those reactions that were not topologically linked to biomass production and, thus, were not predicted to contribute to *Pneumocystis* growth, the primary focus of this study. Furthermore, metabolites that lacked any known association with a metabolic pathway or biological function in the CHEBI database [43] were considered as "orphan" compounds and subsequently removed from the metabolic network. Two orphan reactions were excluded from the model (Table C in S2 File). Finally, we derived five reactions from the databases and manually added them due to their association with corresponding enzymes that are present in the *Pneumocystis* metabolic network.

The resulting draft reconstruction contained 487 reactions (Table D in S2 File).

## Transport reactions

In order to simulate the model under specific environmental conditions, it is necessary to incorporate transport reactions. In this study, we investigated *Pneumocystis* growth in the culture medium proposed by Merali and co-workers (see *Merali's culture medium composition*, below). To achieve this, we compiled a list of all the nutrients present in this culture medium and assumed that *Pneumocystis* could acquire these nutrients from the surrounding environment either through dedicated transporters or through endocytosis, as suggested by genome data [20] (S3 File).

As it is not possible to culture *Pneumocystis in vitro*, the absence of critical experimental data prevented us from imposing constraints on the model's variables to achieve more accurate results using the FBA approach. Nevertheless, we did constrain the biomass flux and the fluxes associated with its coarse-grained constituents (proteins, DNA, RNA, carbohydrates, phospholipids, sterols, cofactors) to positive values, thereby predicting that these components were solely produced and not consumed. In the absence of empirical data, we opted not to constrain external fluxes to prevent the cell from being impeded in excreting by-products.

To automatically fill "metabolic gaps" in the network, we exclusively relied on the *Pneumocystis* taxonomy (4753) to avoid filling "actual metabolic gaps" that reflect the pathogen's dependence on its host, as this may potentially lead to inaccurate predictions.

## FBA algorithm

We applied the FBA approach to study the metabolic pathways of *P. murina* by calculating the fluxes of all reactions in a metabolic network under studied conditions [23]. To this end, we defined an objective function of the biomass flux $v_{BIOMASS}$, and identified a particular flux distribution in the network that maximises the biomass objective function ($v^*_{BIOMASS}$). We formulated our challenge as an optimisation problem with a quadratic objective and affine inequality constraints (a quadratic programming method), and we relaxed the quasi-steady state assumption (QSSA) to obtain an optimal solution in the following way:

$$v^*_{BIOMASS} = \arg\min(-v_{BIOMASS} + \lambda R^2(v))$$

$$lb < v < ub,$$

where $v$ denotes a vector of the metabolic fluxes; $lb$ and $ub$ are lower and upper bounds of the fluxes (we used $lb = -1000$ and $ub = +1000$ where no empirical data were available); $\lambda > 0$ is the relaxation strength (we used $\lambda = 1$); and $R(v) = \|S \times v\|_2$ is the relaxation term with $S$ denoting a stoichiometric matrix (see *Flux Balance Analysis*).

Experimental measurements and model parameters (such as flux capacities) are typically subject to uncertainties. One way to cope with this biological uncertainty is to relax the QSSA by assuming that certain intermediate metabolites may not reach steady state, reducing the

number of constraints and equations. This enables the model to maintain flux to biomass production by adjusting other parts of the network to compensate for missing components. Consequently, the model focuses on overall flux balance rather than precise stoichiometry, allowing for non-zero biomass flux even if certain biomass precursors are missing. QSSA relaxation approach has been successfully applied to a range of GEMs [103–106], notably in the work of Reznik *et al* [107] which demonstrated that flux imbalances could provide valuable information about *in vivo* metabolite concentrations.

## Technical quality of the model

To evaluate the quality and completeness of our GEM, we utilised MEMOTE, a comprehensive metabolic model testing tool [108]. MEMOTE provides a standardised framework for assessing model performance based on various criteria, including stoichiometric consistency, completeness of annotation, and network topology.

These tests revealed high stoichiometric consistency (100.0%), mass balance (94.7%), charge balance (100.0%), and metabolite connectivity (99.8%). In terms of metabolite annotation, the presence was complete at 100.0%, with highly annotated CHEBI (98.6%). Reaction annotations were complete at 100.0%, with high conformity to the Rhea database (96.3%). SBO terms were uniformly present across metabolites, reactions, and genes at 100.0%. Overall, the model achieved a total score of 74%, indicating robust construction with room for improvement in annotation completeness.

The complete MEMOTE report, detailing all assessed criteria and scores, is available in S7 File.

## Merali's culture medium composition

To simulate the model, we used *in vitro* growth culture medium conditions designed by Merali *et al* [16]. These authors reported that continuous axenic culture of *P. carinii* was achieved in the culture medium based on Minimal Essential Medium with Earle's salt supplemented with S-adenosyl-L-methionine, putrescine, ferric pyrophosphate, N-acetyl glucosamine, putrescine, p-aminobenzoic acid, L-cysteine, L-glutamine, and horse serum. The organism grew attached to a collagen-coated porous membrane at normal atmosphere and 31°C with the pH of the medium ranging between 8.8 and 9. Growth was measured by employing three methods, namely, the estimation of total DNA and total ATP, in addition to a conventional microscopic evaluation of Giesma-stained smears. Doubling times varied from 35 to 44 hours in four independent experiments. It was also reported that cultured organisms were able to infect immunosuppressed rats and could be stored frozen and used to reinitiate culture. However, this cultivation method had its limitations. In particular, no growth was observed after dilutions to the point that only 5–10 clusters of *P. carinii* could be observed. There was no growth either in uncoated inserts. Also, the authors reported that medium exchange twice daily was crucial for the growth. Furthermore, other researchers worldwide have reported that their attempts to establish the continuous axenic culture system described by Merali for *P. carinii* met with varying degrees of success. In certain cases, researchers documented their successful attempts in achieving axenic cultivation of *Pneumocystis* [109], while others reported the absence of growth [110].

## Reinforcement Learning algorithm

Reinforcement Learning (RL) is a machine learning paradigm where the algorithm is focused on goal-directed learning via interactions between the agent and the environment [67]. The agent takes actions in the environment and receives feedback in the form of rewards (S1 Fig), which helps it to learn which actions are desirable and which to avoid through trial and error.

The environment with which the agent interacts provides both state and action spaces. The decision of which action to take next is made based on the current state. The policy maps states to actions and determines which action to take in each state. The policy's quality is evaluated and updated based on the rewards received following each step. The objective is to maximise the long-term cumulative reward by balancing the need to attempt new actions to acquire further knowledge about the environment with the need to take actions that will result in the highest reward.

To optimise the growth medium composition, we used this RL approach to find the minimal irreducible subset of nutrients from the initial Merali's 155-nutrient set without dropping the biomass flux value. For this purpose, we treated a subset of nutrients as a state in the state space (the set of all possible subsets), the removal of a specific nutrient as an action, and considered a positive reward if the biomass flux value corresponding to the selected nutrient set was equivalent to that of the previous nutrient set.

An RL routine typically comprises a series of consequent episodes. In our case, an episode was defined as a sequence of nutrient deletions commencing from the initial nutrient set and lasting until no nutrient could be excluded without dropping the biomass flux value.

We chose the Proximal Policy Optimisation (PPO) algorithm [111] as a particular RL method because PPO is widely regarded as a state-of-the-art RL algorithm known for its effectiveness in various domains and it has been implemented in different RL libraries. We employed the implementation provided by StableBaseline3 [112].

## Supporting information

**S1 File. Biomass equation reconstruction.**
(XLSX)

**S2 File. Draft reconstruction.** Table A. Added reactions in the *Pneumocystis* network. Table B. Imbalanced reactions in the *Pneumocystis* network. Table C. Isolates in the *Pneumocystis* network. Table D. Final list of metabolic reactions in the *Pneumocystis* network. Table E. Gene—EC number associations. Table F. Present and lost EC numbers in *P. murina* versus a consensus *S. cerevisiae* metabolic model.
(XLSX)

**S3 File. List of transport reactions under Merali's medium conditions.**
(XLSX)

**S4 File. Merali's medium optimisation.** Table A. List of transport reactions under Merali's medium conditions together with the corresponding flux values. Table B. List of nutrients in the minimal growth medium predicted by the RL approach.
(XLSX)

**S5 File. Merali's medium optimisation. Table A. Model predictions.** Table B. Flux values under GLUCOSE conditions. Table C. Flux values under MERALI conditions. Table D. Flux values under MERALI + LIPIDS conditions.
(XLSX)

**S6 File. Metabolic model of *Pneumocystis murina*.**
(XML)

**S7 File. MEMOTE report for the metabolic model of *Pneumocystis murina*.**
(HTML)

**S1 Fig. Schematic representation of the RL approach used in this study.** RL algorithms operate on states, actions, and rewards within an environment. The RL agent decides of which action to take based on the current state. The policy maps states to actions and is updated according to the obtained reward. The objective is to maximise cumulative reward by balancing exploration and high-reward actions.
(TIF)

**S2 Fig. Learning process of the RL approach used in this study. (A)** The learning process of identifying and eliminating non-essential nutrients from the initial list of 155 nutrients is depicted by the blue and orange curves, which represent the precise and average sizes of the nutrient set after 118 attempts to remove nutrients by episodes. The value 118 represents the minimum number of nutrient deletions necessary to reduce the initial set of 155 nutrients to a minimal set consisting of 37 nutrients. **(B)** The distribution of the steps on which attempts made to eliminate choline, a non-essential nutrient (depicted on the top panel), and cholesterol, an essential nutrient (depicted on the bottom panel), from the initial list of 155 nutrients in Merali's medium during each episode. The figures indicate that during the learning process the essentiality of cholesterol has been established, suggesting that it should not be removed at the onset of the episode. On the other hand, since choline is non-essential, it is advisable to eliminate it during the middle phase of the episode.
(TIF)

# Acknowledgments

The authors would like to thank the research collaborators who generously contributed their expertise and knowledge to this project, including Prof Jay K. Kolls, Dr Thibault Etienne, Benjamin Maisonneuve, and Maëva Beugin for helpful discussions. Additional work may have been undertaken by the University of Exeter Biological Services Unit.

# Author Contributions

**Conceptualization:** Olga A. Nev, Alistair J. P. Brown.

**Formal analysis:** Olga A. Nev, Elena Zamaraeva.

**Funding acquisition:** Olga A. Nev, Elena Zamaraeva, Lucian Duvenage, Jennifer Claire Hoving, Ivana Gudelj, Alistair J. P. Brown.

**Investigation:** Lucian Duvenage, Jennifer Claire Hoving.

**Methodology:** Olga A. Nev, Wassim Abou-Jaoudé, Djomangan Adama Ouattara.

**Resources:** Lucian Duvenage, Wassim Abou-Jaoudé, Djomangan Adama Ouattara, Jennifer Claire Hoving.

**Software:** Olga A. Nev, Elena Zamaraeva, Romain De Oliveira, Ilia Ryzhkov, Wassim Abou-Jaoudé, Djomangan Adama Ouattara.

**Supervision:** Olga A. Nev, Alistair J. P. Brown.

**Writing – original draft:** Olga A. Nev, Elena Zamaraeva, Romain De Oliveira, Wassim Abou-Jaoudé, Djomangan Adama Ouattara, Jennifer Claire Hoving, Alistair J. P. Brown.

**Writing – review & editing:** Olga A. Nev, Elena Zamaraeva, Ilia Ryzhkov, Wassim Abou-Jaoudé, Djomangan Adama Ouattara, Alistair J. P. Brown.

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
