## [Decision Letter · Decision Letter 0]

28 May 2024

Dear Dr Nev,

Thank you very much for submitting your manuscript "METABOLIC MODELLING AS A POWERFUL TOOL TO IDENTIFY CRITICAL COMPONENTS OF PNEUMOCYSTIS GROWTH MEDIUM" for consideration at PLOS Computational Biology.

As with all papers reviewed by the journal, your manuscript was reviewed by members of the editorial board and by several independent reviewers. In light of the reviews (below this email), we would like to invite the resubmission of a significantly-revised version that takes into account the reviewers' comments. Please pay particular attention to the points of the reviewers concerning clarification of the the methods you used and making source code as well as the model available in SBML format.

We cannot make any decision about publication until we have seen the revised manuscript and your response to the reviewers' comments. Your revised manuscript is also likely to be sent to reviewers for further evaluation.

Sincerely,

Christoph Kaleta

Section Editor

PLOS Computational Biology

Stacey Finley

Section Editor

PLOS Computational Biology

Reviewer's Responses to Questions

**Comments to the Authors:**

Reviewer #1: The manuscript "METABOLIC MODELLING AS A POWERFUL TOOL TO IDENTIFY CRITICAL COMPONENTS OF PNEUMOCYSTIS GROWTH MEDIUM" by Nev and colleagues presents the reconstruction of a genome-scale metabolic model of the fungal pathogen *Pneumocystis jirovecii*. In general, the reconstruction of metabolic networks in fungi remains a major challenge for systems biology and more efforts such as the present study are needed. However, I do have a few critical points about the manuscript, which mainly refer to data availability, quality control of the genome-scale model, and results interpretation. Unfortunately, in the present manuscript version it is partly not comprehensible how the authors came to certain statements (e.g. about amino acid auxotrophies), because the corresponding results are not shown in the manuscript.

Introduction, 3rd paragraph: "we remain remarkably ignorant about the biology ... of this pathogen" is a harsh expression (also in the abstract). I am sure that researchers that study Pneumocystis jirovecii would disagree because the organism has been studied before using culture-independent methods, which also provide insights into the biology of the pathogen.

The abstract and introduction build on the central statement that P. jirovecii is not culturable. Yet, the last paragraph of the introduction cites a paper by Merali et al., where the organism was growth under laboratory conditions for a limited duration of time. Since this particular paper seems crucial in the context of the present manuscript, this paper needs to be introduced in greater detail already in the introduction. This should address the questions: What is exactly meant by "limited duration"? What was the innovation in the study that supported P. j. growth? What remains unclear after the Merali study and why is genome-scale modelling the method of choice to address open questions?

Section METABOLIC MODEL DEVELOPMENT: "Pneumocystis [...] is incapable of synthesising all 20 essential amino acids". That seems very unlikely. For instance, there are plenty of different metabolic routes to produce alanine, aspartate, and glutamate, so that established auxotrophy prediction tools like "GapMind" do not make statements about auxotrophies for these amino acids. Even more importantly: It is unclear how auxotrophies were assessed or predicted. What precursors/pathways were considered to make the statement of the incapability to synthesise all 20 proteinogenic amino acids. Moreover, Figure 7A clearly shows that the model can produce biomass without certain amino acids in the growth medium, indicating that the organism is capable of synthesizing at least some amino acids.

Biomass reaction. It is mentioned that the biomass reaction was defined based on ""genome data, software tools, pre-existing metabolic models of species closely related to Pneumocystis, and the limited information about the pathogen’s composition available in the literature". However, the described procedure misses essential information to make the biomass reaction definition reproducible. For instance "the aid of a java application" is too vague and more details need to be provided (what is this application, what are underlying assumptions and data sources?). The same applies to this part: "majority of the content for the soluble pool of small metabolites was derived from a comprehensive general list". Please provide more details on this "general list".

There are no results on the technical quality of the reconstructed GEM. This could for instance be done using MEMOTE (https://doi.org/10.1038/s41587-020-0446-y).

Optimization of Merali's medium. I can't find a table that represents the reconstructed Merali's medium, which is used as reference for several analysis presented in the manuscript.

The metabolic model should be provided in SBML format also already for the review process, since this is the main result of the study.

Please add line numbers. This makes reviewing easier.

Reviewer #2: Nev and colleagues use metabolic modelling to identify necessary components of the fungal pathogen Pneumocystis growth medium. P. jirovecii is an important pathogen that cause lethal pneumonia in immunocompromised patients. The fungus is difficult to study due to a lack of long-term culture method. Here, the authors leverage on available genomic data, insights from more than 30 years of culture attempts to model the metabolic fluxes in P. murina, the sister species of P. jirovecii.

This study addresses a need in the research community. The manuscript is well written, the methods are generally appropriate, and the conclusions are supported by the results provided. The comments below are provided to discuss some concepts and potentially improve the manuscript.

This study is a proof of concept showing that metabolic modelling can help achieving a reliable continuous culture of Pneumocystis organisms in an axenic setting. One of the main predictions of the model is Pneumocystis reliance on glucose. This is hardly new and has already been inferred from Pneumocystis gene catalogs (See Ma et al. 2016 for example). It is my understanding that the authors took the safe route and showed only predictions that fit the current narrative. It should be noted that the link between gene loss and metabolic dependency is far from being proven. As is the model seeming under-utilized. Can the authors present or emphasize on new or counterintuitive results not predicted in the literature?

I understand that answering this question can be risky because this paper is a purely in silico study, and producing predictions without experimental backup can be hard to defend. However, it would be of interest to to Plos Comp Biol readers.

The reliance on Merali et al paper to confirm their predictions is a major - but easy to fix - drawback. The method described in this paper does not work (even mentioned by the authors). Efforts to replicate this study have failed. Melanie Cushion et al (ref #16) has nicely summarized efforts done in supplementing the media. Therefore, the reliance of Merali is not warranted.

The other argument is Pneumocystis is an obligate biotroph that apparently need a host. With that in mind, axenic methods are unlikely to work and co-culture methods are more successful (PMID: 38117035). This aspect is not captured by this modelling. The effect is exacerbated when an inappropriate dataset is used such as Merali. Can the authors please comment>

Finally, adding elements to growth media is not enough. And many of these elements have been tested already (see ref # 16).

Specific comment

In particular, we estimated Pneumocystis protein and DNA content from genomic data with

the aid of a java application [37].

What the tool is doing exactly?

“FBA assumes that the metabolic system is in a quasi-steady state, where metabolite

production and consumption are balanced within the cell.”

Can public RNA-Seq data be used to initialize the model? For example, gene expression can be used to capture the cellular state in different conditions (e.g. ref 32). Where are the codes (or GitHub repo)?

“To ensure that the model is free of technical errors, the model is

usually tested against known constraints. To test whether the model accurately represents

the metabolism of the organism, the predictions of the model are compared with

experimental data obtained under various environmental conditions.”

This sentence is unclear. There are no environmental conditions available. Also, culture system does not recapitulate in vivo conditions.

“data are limited and often obtained under ex vivo conditions (i.e. growth with cultured

mammalian cells) which are not suitable for the model’s validation. To validate our

metabolic model, we employed in vitro growth culture medium conditions designed by

Merali et al. [30]”

As discussed, Merali et al work is not appropriate.

“conditions and discovered 91 zero fluxes associated with extracellular transport reactions (SI

4, Table S5). These fluxes do not appear to contribute to biomass production and, therefore,

the corresponding ambient nutrients from the medium can be excluded. We compared the

nutrients identified as non-essential for growth by the RL approach with nutrients whose

corresponding flux values were zero. The RL algorithm identified a significantly greater

number of non-essential nutrients, considerably simplifying the original 155-component list

of Merali’s medium composition to only 37 nutrients.”

This whole paragraph is about predicting non-essential genes. Can RL be trained on RNA-seq data? Or what would be the outcomes if RL is trained on different datasets?

“Merali et al did not experimentally detect any activity of the SAM synthase and thus

suggested that P. carinii is a SAM auxotroph … suggest that Pneumocystis may not depend on its host for the supply of this important cofactor.”

This paragraph has an historical value and would be appropriate for a review, but here it is confusing. Pneumocystis organisms definitively have a SAM synthase. This reinforces my suspicion that Merali et al work is not reliable.

“We have developed, for the first time, a genome-scale metabolic model for Pneumocystis. In”

Efforts to model metabolic pathways have been attempted before (for example Porollo 2014, PMID: 25202338; Hauser et al. PMID: 21188143)

Reviewer #3: The here presented manuscript presents a genome-scale metabolic reconstruction of Pneumocystis murina, a representative of the Pneumocystis genus.

The authors describe reconstruction details, FBA based simulations and model predictions. One major drawback of Pneumocystis research appears to be the inability to have it grown in vitro so far. Hence, a genome-scale metabolic model (GEM) to shed some light into its metabolic components features appears timely.

Although the limited knowledge about Pneumocystis biology hindered some parts of the reconstruction and FBA-based analysis, I have major concerns regarding the methodology and miss supplementary information with model and simulation details. Moreover, the predicted minimum growth medium was not experimentally tested and thus remains hypothetical.

Major points

--

In the model predictions section, how was influx modeled? Were all influx exchange-reaction-related bounds open or in any way constrained to specific flux values e.g. as given by the definition of the Merali medium? This has definitely impact on model predictions and needs to be stated here. Also, a complete Suppl. table with model predictions (and used influx values and achieved FBA objective function values as well as all model flux values) is missing.

- Moreover, since Arginine biosysythesis is not complete as shown in Fig 2B, how do the authors explain that missing arginine in the medium has such low impact? It does not matter, for predicted viability (FBA>0) to which extend an amino acid (or any other factor) contributes to the biomass equation. If it is essential, FBA is necessarily 0. Obviously the individual lack of each amino acid could be compensated as shown in Fig. 6. How, is not explained. The authors need to address how the missing amino acids are produced by investigating the respective FBA fluxes or even better by running reaction essentiality analysis with vs. without a given amino acid using e.g. FVA.

The section on glycine/serine and SHMT is unclear. A gene KO has no effect, but if the reaction is knocked out, the authors report an effect (no serine-glycine interconversion). The authors need to explain this, e.g. has the respective gene-to-reaction rule an isozyme taking over the role for enabling the reaction? Otherwise it is not clear how the reaction was functioning after the gene-KO. The effect should have been the same as seen by the predicted flux after reaction-KO. Also, I miss gene-to-reaction rules in the supplements for each reaction and the evidences on which basis reactions were added to the final model. Comprehensive information which reaction associates to which metabolic pathway should also be given.

When investigating the role of lipids the authors again need to explain how influx was modeled (constrained or not to certain values). Also again, if any component is part of the biomass composition, its prohibitited influx should result in FBA=0, unless the compound is produced elsewhere.

The paper would benefit if the authors would use their model to predict the reactions necessary to produce certain sterols and fatty acids, which were added to the biomass reaction based on information in references 38-40, but are not part of the growth medium by Merali. This can only mean the model is able to synthesize these, which is not necessary if additionally given in the medium - as has been simulated and shown in Fig. 5. On its own this is a trivial result. But if the authors add predictions/simulations on which pathways are involved in the biosynthesis and e.g. what energy demands are circumvented by supplementing the respective metabolites to the medium, the advantage of having the model at hand would be demonstrated.

Concerning the growth medium optimisation, it is certainly interesting to see this great reduction in necessary metabolites from >800 to 37. Toolboxes like CobraPy allow to predict a minimal medium, which could be exploited to find all possible variants with a minimal number of metabolites. How does the presented RL based approach compare to this rather straight forward approach included in CobraPy? Are the core nutrients found also in there?

Table S5 presents 91 zero fluxes of given medium components. Does this imply these are not taken up by Pneumocystis at all or are these values associated to the minimum already? How do the authors explain that e.g. L-argininium(1+) is a core metabolite after RL application according to S6, but the respective flux in S5 is 0? How should negative and positive fluxes in S5 be interpreted, since in this table they reflect the Merali medium, for which there should be only uptake or zero flux?

How does metabolomics data (if avialable) before and after Pneumocystis growth compare to these presented values?

In the methods section, the quadratic objective instead of regular FBA is insufficiently explained and justified. Why has the presented formular been adapted instead of regular FBA? Why have relaxation terms been added to the problem formulation? Has this been used and published elsewhere? What does "see FBA analysis" refer to as there is no such section?

The model should be tested with MEMOTE and should be available in SBML format.

Minor points

--

Introduction misses a description of existing fungal genome-scale metabolic models.

Table 1 is potentially interesting for follow-up studies. It misses the involved lost or retained metabolic reactions, however, and I expect these to see in a Supplementary Table.

The section "Biomass equation" would benefit from a reference to pmid20430689 for further information on biomass objective function.

The first paragraph on FBA lists some FBA application papers, the youngest being published in 2012. Please add 1-3 more recent applications.

The authors repeat the understandable limitation of Pneumocystis not growing in culture and therefore data for validation is scarce. Nevertheless, the model validation section needs more explanation, mainly to describe what the authors did (or not) to remedy the discrepancy between observed and simulated growth.

Next to confirming observations by Merali and others which metabolites appear important, did the employed approach also predict essential nutrients that so far were not mentioned in the literature?

The discussion mentions for the first time clearly that a GEM model of P. murina was reconstructed. How different or similar is this related to e.g. P. jirovecii? How much are genome annotations differing, which potential metabolically relevant genes are present in one, several or all? What about homology-searches of the organism-specific Pneumocystis species against e.g. S. cerevisiae genome annotation and comparison against the Yeast consensus model, which is one of the best curated ones? The understandable reasons why to reconstruct a P. murina model as starting point and to resemble the Pneumocystis genus should be made clear in the very beginning of the results. Also, throughout the paper, instead of describing "metabolic model for Pneumocystis" or "critical components of the Pneumocystis growth", etc. it should be made clear that either P. murina or the genus is meant.

In the materials and methods section, the description "Finally we checked the obtained list of enzymes against the biochemical reaction database BRENDA [34] and RNA Seq data [32], and also used available, albeit limited, experimental data on Pneumocystis to curate enzymes manually." appears not precise enough. How was this done. The supplements with the affected reactions should be referenced here.

Are the gene-to-reaction rules derived by the ConstellabTM software tool different or equal for the same reactions present also in the yeast consensus model (https://github.com/SysBioChalmers/yeast-GEM)? A comparison would improve trust in the given model, which is otherwise based on proprietary black-box software.

The deleted "isolate" reactions may not be relevant for growth, but this largely depends on the given growth medium. As the authors repeatedly state, stable in vitro growth so far is not possible and only the Merali medium supports growth for some time. This, however, does not rule out other possible growth media compositions, for which one or several of the deleted reactions may become relevant. Even if not, these may be important for pathogenicity and thus, helpful to other researchers and the community. At the minimum, their reaction details should be added and described in the supplements, e.g. in SI2.

s

The gap filling procedures need to be elaborated. Is this step included in the pipeline by the Constellab software? Since viability of the model was achieved, the question arises, which and how many gaps were filled given which resource. Such information should be in the per reaction supplementary tables, e.g. in table S4.

In the methods description of Merali’s culture medium composition, the culture medium appears to be based on P. carinii, however, the model built here was P. murina. The text should reflect again on the similarity (or not) of these strains.

**Have the authors made all data and (if applicable) computational code underlying the findings in their manuscript fully available?**

Reviewer #1: **No: **The genome-scale metabolic model should be provided in SBML format. Also model analysis code and central data tables (e.g. the merali growth medium used for FBA) should be made available.

Reviewer #2: **No: **processed data are provided, but codes are not. Please consider a Github repo

Reviewer #3: **No: **Code for model reconstruction and simulation are not available. The model itself is not available in common SBML format.

PLOS authors have the option to publish the peer review history of their article (what does this mean?). If published, this will include your full peer review and any attached files.

Reviewer #1: No

Reviewer #2: No

Reviewer #3: No
---

## [Decision Letter · Decision Letter 1]

19 Aug 2024

Dear Dr Nev,

Thank you very much for submitting your manuscript "METABOLIC MODELLING AS A POWERFUL TOOL TO IDENTIFY CRITICAL COMPONENTS OF PNEUMOCYSTIS GROWTH MEDIUM" for consideration at PLOS Computational Biology. As with all papers reviewed by the journal, your manuscript was reviewed by members of the editorial board and by several independent reviewers. The reviewers appreciated the attention to an important topic. Based on the reviews, we are likely to accept this manuscript for publication, providing that you modify the manuscript according to the review recommendations.

Sincerely,

Alison Marsden

Section Editor

PLOS Computational Biology

Stacey Finley

Section Editor

PLOS Computational Biology

Reviewer's Responses to Questions

**Comments to the Authors:**

Reviewer #1: The revised manuscript demonstrates several improvements, including the inclusion of the metabolic model in SBML format. However, two of my main concerns from the previous report remain unresolved:

Auxotrophies: In my previous report, I raised concerns regarding the predictions of amino acid auxotrophy. While the authors have expanded the discussion on amino acid auxotrophies, I remain unconvinced by the current response. The manuscript and response letter still assert that *Pneumocystis jirovecii* is unable to produce any of the 20 proteinogenic amino acids. This assertion contradicts the results presented in the section "In silico Drop-off Experiments" and in Figure 6. In this section, the manuscript describes the results as follows:

"The results suggest that not all amino acids contribute equally to Pneumocystis growth. In particular, the model predicts that a deficiency of isoleucine or histidine in the medium results in a significant reduction in biomass flux, whereas arginine or tyrosine deficiencies do not (Fig 6)."

If the organism is indeed auxotrophic for all 20 amino acids, the maximum flux through the biomass reaction should be reduced to zero when the organism is unable to synthesize the respective amino acids. However, Figure 6 clearly shows that the relative biomass flux remains greater than zero for all amino acids tested through exclusion from the in-silico growth medium. This inconsistency needs to be addressed to clarify the organism's metabolic capabilities.

Biomass Reaction: Although the authors have added further details on how the biomass reaction was inferred, the information provided is still incomplete, which hinders the reproducibility of the biomass formulation in the metabolic model. Specifically, the revised manuscript states:

"In particular, we estimated Pneumocystis protein and DNA content from genomic data with the aid of a biomass tool developed by Santos & Rocha [49]. This tool calculates the microbial biomass composition in amino acids and nucleotides based on genomic and transcriptomic data, accepting input files in FASTA format and transcriptomic data in CSV format. It is reported to serve as a viable alternative in the absence of experimental data [49-51]"

To ensure reproducibility, it is essential to provide the version number of the Java tool used and to specify which transcriptomic data was employed as input. This additional information will help others in the field to accurately reproduce the biomass formulation.

Reviewer #2: The authors have resolved many technical issues and improved the overall quality of the manuscript.

Minor point:

“It seems that nutrient supplementation alone is not sufficient to sustain growth. Instead, the organism must be able to maintain intracellular energetic and redox balance, for example. Therefore, to address this, we used metabolic modelling as an alternative approach (lines 94-100).”.

Is there a reference for this statement? The growth issue is unclear. There is a distinction between short term limited growth, which several groups have achieved (see Cushion et al. ref # 19) and long-term growth with passages, which has been impossible to achieve so far. This point should be clearly explained in the introduction.

Reviewer #3: I thank the authors for their serious effort to provide additional information and to address the reviewers' comments.

I find the manuscript overall strengthened.

My main concerns around predictions made with FBA were already addressed by modifying the ILP problem as a quadratic optimization problem. It is my humble opinion that this misunderstanding can easily happen to others in the community, as the main text and figure 4 describe FBA and its steady state assumption without any adaptation, which led to the points I raised. The authors sufficiently improved the description in the methods section to make this clear.

I kindly request the authors to clarify and strengthen their description also in the "Flux Balance Analysis" section of the main text and Figure 4 (which still describes classical FBA). For instance the authors write "FBA assumes that the metabolic system is in a quasi-steady state, where metabolite production and consumption are balanced within the cell." This was misleading and strictly speaking this is not true, as FBA on its own assumes a steady-state of all systems components. In the current work this was adapted to a quasi-steady state and relaxed bounds requirements (as for example termed "relaxed flux balance analysis" by Fleming et al 2023 in reference 105) to allow for feasible model simulation and subsequent simulation analyses. Excuse me if this may appear nitpicking, I believe the differentiation is important and helps the manuscript if clarified.

In any case, the confusion with strict FBA (and its Sv=0 requirements for all model components) can be easily avoided, if the authors reformulate relevant parts in the Flux Balance Analysis section and update Figure 4 to clarify that they used an adapted FBA approach to relax bounds with a quasi-steady state as detailed out in the methods section.

**Have the authors made all data and (if applicable) computational code underlying the findings in their manuscript fully available?**

Reviewer #1: **No: **The "DATA AVAILABILITY" section mentions github repositories of software tools that the authors have used for their analysis. The code, that the authors used for their own analysis (metabolic network reconstruction and anaylsis), is not provided.

Reviewer #2: Yes

Reviewer #3: None

PLOS authors have the option to publish the peer review history of their article (what does this mean?). If published, this will include your full peer review and any attached files.

Reviewer #1: No

Reviewer #2: No

Reviewer #3: No

Figure Files:

Data Requirements:

Reproducibility:

References:

---

## [Decision Letter · Decision Letter 2]

16 Sep 2024

Dear Dr Nev,

Thank you very much for submitting your manuscript "METABOLIC MODELLING AS A POWERFUL TOOL TO IDENTIFY CRITICAL COMPONENTS OF PNEUMOCYSTIS GROWTH MEDIUM" for consideration at PLOS Computational Biology. As with all papers reviewed by the journal, your manuscript was reviewed by members of the editorial board and by several independent reviewers. The reviewers appreciated the attention to an important topic. Based on the reviews, we are likely to accept this manuscript for publication, but I'd like to ask you to clarify the one remaining point of reviewer 1 and the point concerning code availability raised by reviewer 3 before the manuscript can be accepted.

Sincerely,

Christoph Kaleta

Section Editor

PLOS Computational Biology

Stacey Finley

Section Editor

PLOS Computational Biology

Reviewer's Responses to Questions

**Comments to the Authors:**

Reviewer #1: I appreciate the detailed clarification. With the modifications in the manuscript, it is now more clearly illustrated that the presented results are specific to the assumption that fluxes are not constrained by strict steady-state conditions.

Regarding the authors' response, they stated: "we do not say that Pneumocystis is auxotrophic for all 20 amino acids." However, the manuscript contains the following sentence: "[...] Pneumocystis has partially lost the gluconeogenesis pathway and glyoxylate cycle and is incapable of synthesizing all 20 essential amino acids de novo [...]". These statements appear to be contradictory.

I understand that, due to the quasi-steady state assumption (QSSA) approach, auxotrophies cannot be predicted based solely on zero flux values for biomass formation when a specific amino acid is removed. However, given that the aim of the study is to contribute to the identification of a medium that supports Pneumocystis growth, readers might reasonably expect a discussion on the results in relation to auxotrophies. In the current version of the manuscript, the focus is on the contribution of amino acids to biomass production, which does not provide new insights regarding potential auxotrophies. If the chosen approach (QSSA-FBA) does not allow for auxotrophy prediction, it would be helpful to clearly state this as a limitation in the discussion.

Reviewer #2: The authors have adressed most of the concerns.

Reviewer #3: My remaining comments have been addressed by the authors. The differentiation between relaxed FBA and classic FBA is sufficiently clear now.

**Have the authors made all data and (if applicable) computational code underlying the findings in their manuscript fully available?**

Reviewer #1: Yes

Reviewer #2: Yes

Reviewer #3: **No: **The authors provide the reconstructed SBML model as supplementary file.

Provided links refer to the proprietary software workbench Constellab and the description of generating a workflow.

However, custom analysis code that led to the presented results are not available.

The mentioned "digital lab" description should be made available.

PLOS authors have the option to publish the peer review history of their article (what does this mean?). If published, this will include your full peer review and any attached files.

Reviewer #1: No

Reviewer #2: No

Reviewer #3: No

Figure Files:

Data Requirements:

Reproducibility:

References:

---

## [Editor Report · Decision Letter 3]

9 Oct 2024

Dear Dr Nev,

We are pleased to inform you that your manuscript 'METABOLIC MODELLING AS A POWERFUL TOOL TO IDENTIFY CRITICAL COMPONENTS OF PNEUMOCYSTIS GROWTH MEDIUM' has been provisionally accepted for publication in PLOS Computational Biology.

Best regards,

Christoph Kaleta

Section Editor

PLOS Computational Biology

Stacey Finley

Section Editor

PLOS Computational Biology

---

## [Editor Report · Acceptance letter]

18 Oct 2024

PCOMPBIOL-D-24-00648R3 

Metabolic modelling as a powerful tool to identify critical components of Pneumocystis growth medium

Dear Dr Nev,

I am pleased to inform you that your manuscript has been formally accepted for publication in PLOS Computational Biology. Your manuscript is now with our production department and you will be notified of the publication date in due course.

With kind regards,

Anita Estes
